# Retrieval of Terahertz Ice Cloud Properties from airborne measurements based on the irregularly shaped Voronoi ice scattering models

## Ming Li[1,2], Husi Letu[1], Hiroshi Ishimoto[3], Shulei Li[4], Lei Liu[4], Takashi Y. Nakajima[5], Dabin Ji[1], Huazhe Shang[1], Chong Shi[1]

[1]State Key Laboratory of Remote Sensing Science, Aerospace Information Research Institute, Chinese Academy of Sciences, Beijing, 100101, China
[2]University of Chinese Academy of Sciences, Beijing, 100049, China
[3]Meteorological Research Institute, Japan Meteorological Agency (JMA), Nagamine 1-1, Tsukuba, 305-0052, Japan
[4]College of Meteorology and Oceanography, National University of Defense Technology, Changsha, 410073, China
[5]Research and Information Center (TRIC), Tokai University, 4-1-1 Kitakaname Hiratsuka, Kanagawa, 259-1292, Japan

*Correspondence to*: Husi Letu (husiletuw@hotmail.com)

**Abstract.** Currently, terahertz remote sensing technology is one of the best ways to detect the microphysical properties of ice clouds. Influenced by the representativeness of the ice crystal scattering (ICS) model, the existing terahertz ice cloud remote sensing inversion algorithms still have significant uncertainties. In this study, based on the Voronoi ICS model, we developed a terahertz remote sensing inversion algorithm of the ice water path (IWP) and median mass diameter ($D_{me}$) of ice clouds. This study utilized the single-scattering properties (extinction efficiency, single-scattering albedo and asymmetry factor) of the Voronoi, Sphere and Hexagonal Column ICS models in the terahertz region. Combined with 14,408 groups of particle size distributions obtained from aircraft-based measurements, we developed the Voronoi, Sphere and Column ICS schemes based on the Voronoi, Sphere and Column ICS models. The three schemes were applied to the RSTAR radiative transfer model to carry out the sensitivity analysis of the top of cloud (TOC) terahertz brightness temperature differences between the cloudy and clear sky (BTDs) on the IWP and $D_{me}$. The sensitivity results showed that the TOC BTDs between 640 and 874 GHz are functions of the IWP, and the TOC BTDs between 380, 640 and 874 GHz are functions of the $D_{me}$. The Voronoi ICS scheme possesses stronger sensitivity to the $D_{me}$ than the Sphere and Column ICS scheme. Based on the sensitivity results, we built a multi-channel look-up table for BTDs. The IWP and $D_{me}$ were searched from the look-up table using an optimal estimation algorithm. We used 2000 BTD test data randomly generated by the RSTAR model to assess the algorithm's accuracy. Test results showed the correlation coefficients of the retrieved IWP and $D_{me}$ reached 0.99 and 0.98, respectively. As an application, we used the inversion algorithm to retrieve the ice cloud IWP and $D_{me}$ based on the CoSSIR airborne terahertz radiation measurements. Validation against the retrievals of the Bayesian algorithm reveals that the Voronoi ICS model performs better than the Sphere and Hexagonal Column ICS models, with the enhancement of the mean absolute errors of 5.0% and 12.8% for IWP and $D_{me}$, respectively. In summary, the results of this study confirmed the

practicality and effectiveness of the Voronoi ICS model in the terahertz remote sensing inversion of ice cloud microphysical properties.

## 1 Introduction

Ice clouds account for about 20-30% of the total global cloud mass (Liou, 1992; Rossow and Schiffer, 1991). They play an essential regulatory role in the global radiation balance and the water cycle (Hong et al., 2016; Stephens et al., 2012). Microphysical properties such as the ice water content, ice particle size, orientation and shape are the main influencing factors of the scattering and radiative properties of ice clouds (Li et al., 2022; Yi et al., 2017; Zhao et al., 2018; Chen and Zhang, 2018) and, in turn, affect the radiation budget (Heymsfield et al., 2013b, 2017; Rossow, 2014). The latest report of

the 6th Intergovernmental Panel on Climate Change (IPCC6) (Forster et al., 2021) identifies cloud radiative properties and their feedback effects as the largest source of uncertainty in the overall climate feedback, with errors in ice clouds being one of the most significant factors (Zhang et al., 2021). Therefore, the accurate acquisition of microphysical properties of ice clouds is of great importance for studying global climate change and improving the accuracy of numerical model simulations (Baran, 2009, 2012). Remote sensing techniques are one of the most effective means of obtaining microphysical and

radiative properties of ice clouds, mainly including ground-based (Cimini et al., 2006), aircraft-based (Fox et al., 2017) and satellite-based remote sensing observation technologies (Yang et al., 2015, 2018). Currently, large amounts of passive sensors (visible, infrared and microwave detectors) have been developed, and related ice cloud retrieval algorithms have been reported in substantial literature (Nakajima and King, 1990; Nakajima et al., 1991, 2019; Nakajima and Nakajima, 1995; Platnick et al., 2003, 2017; Fox et al., 2019; Brath et al., 2018). The visible and infrared wavelengths are sensitive to the

visible optical depth and cloud top (Minnis et al., 1993b; Minnis et al., 1993a). The millimetre-wave ice cloud remote sensing technique is more suited to detect vertical cloud properties. Sensors such as the Millimeter-wave Imaging Radiometer (MIR) (Racette et al., 1992) and Special Sensor Microwave Water Vapor Sounder (SSM/T-2) have been used in several studies of IWP and particle size retrievals (Lin and Rossow, 1996; Liu and Curry, 1998, 1999). MIR channels at 89, 150 and 220 GHz have been used by Deeter and Evans (2000) and Liu and Curry (2000) to retrieve IWP and particle size in

cirrus anvils over the tropical ocean. Compared to passive sensors, the Cloud satellite radar (CloudSat), with an onboard millimetre-wavelength (94.05 GHz) radar and the raDAR/liDAR cloud product (DARDAR) (Ceccaldi et al., 2013) present new opportunities to infer the microphysical properties of ice clouds on a global scale. In practice, the current detectors and approaches are confined to a limited range of ice particle sizes (Cho et al., 2015). For example, visible and infrared sensors are only sensitive to small particles smaller than 50 μm (Evans et al., 2005). Additionally, microwave detectors are mainly

useful for large particles larger than 500 μm (Fox, 2020; Fox et al., 2019). There is a pressing need to develop an effective frequency region for obtaining comprehensive information about ice particles. To bridge the gap between the technologies of visible/infrared and microwave measurement of ice clouds, the terahertz (THz) wavelength between the infrared and

microwave regions has the potential advantage of complementing existing visible/infrared and microwave techniques (Gasiewski, 1992).

The terahertz wave is the submillimeter-wave with frequencies in the range of 0.1~10 THz and corresponding wavelengths of 0.03~3mm, comparable to the size of ice particles in ice clouds. Many theoretical studies (Evans and Stephens, 1995a, 1995b; Evans et al., 1998) have shown that the passive terahertz wave has a higher detection capability and sensitivity to the ice water path (IWP) and particle size of ice clouds (Liu et al., 2020). Although terahertz waves were proposed for ice cloud remote sensing in the 1960s (Gao et al., 2016), the technology of terahertz radiometry of ice clouds lagged behind the theory

(Evans et al., 2005). It is only within the last decade that terahertz radiometry has been applied to aircraft-based and satellite-based ice cloud remote sensing. With advances in terahertz detectors, researchers have successively developed aircraft-based terahertz radiometers (Gao et al., 2016), such as the Sub-millimeter Wave Cloud Ice Radiometer (SWCIR) (Evans et al., 2002), the Compact Scanning Sub-millimeter wave Imaging Radiometer (CoSSIR) (Evans et al., 2012) and the International Sub-Millimeter wave Airborne Radiometer (ISMAR) (Fox et al., 2017). Also, research institutions have developed satellite-

based terahertz radiometers, including Superconducting subMillimeter-wave Limb Emission Sounder (SMILES) (Inatani et al., 2000), Ice Cloud Imager (ICI) (Eriksson et al., 2020; Kangas et al., 2014) and IceCube (Wu et al., 2014), which were proposed in 2013 and are still under development.

Several methods have been reported to retrieve the ice clouds' IWP and particle size from terahertz brightness temperature. Inversion methods can be divided into physical and linear regression methods (Weng et al., 2019). The linear regression

method is efficient and convenient, but it lacks a definite physical mechanism and is heavily dependent on the accuracy of the pre-calculated database. The physical method includes the Bayesian algorithm (Evans et al., 2005; Evans et al., 2002), the look-up table (LUT) method (Li et al., 2017; Li et al., 2018; Liu et al., 2021) and neural network algorithm (Jimenez et al., 2003). Evans et al. (1998) modeled terahertz brightness temperature using a polarized radiative transfer model based on eight ice particle shapes calculated by the discrete dipole approximation (DDA) method. The study found that the ice particle

shape plays a vital role in modeling ice cloud scattering in the terahertz region. Evans et al. (2002) developed a Monte Carlo Bayesian Integration (MCBI) algorithm to retrieve ice clouds' IWP and median mass diameter ($D_{me}$) from simulated SWCIR brightness temperatures. According to the validation results, the total median retrieval error for clouds with IWP of more than 5 g/m$^2$ is roughly 30% for IWP and 15% for $D_{me}$ (Evans et al., 2002). Moreover, Evans et al. (2005) utilized the MCBI method to retrieve the IWP and $D_{me}$ of ice clouds from the CoSSIR brightness temperatures (referred to as the CoSSIR-

MCBI hereafter). During the retrieval procedure, one of five particle shapes (spherical snow, aggregates of frozen droplets, aggregates of hexagonal plates, and aggregates of plates and hexagonal columns) developed by the DDA method (Evans and Stephens, 1995a) was selected for each ice cloud retrieval. The CoSSIR-MCBI results are validated by the Cloud Radar System data and showed a good agreement of radar backscattering with errors smaller than 5 dB. Later, Jimenez et al. (2007) used the neural network method combined with the radiative transfer code to retrieve the IWP and $D_{me}$ of ice clouds. The ice

cloud microphysical input is based on one of three randomly oriented particle shapes (solid columns, hexagonal plates, and four-bullet rosettes) simulated with the DDA method (Evans and Stephens, 1995a; Evans et al., 1998) in the microwave

region. Results showed overall median relative errors of around 20% for IWP and 33 μm for $D_{me}$ for a mid-latitude winter scenario and 17% for IWP and 30 μm for $D_{me}$ for a tropical scenario. Based on early studies and the background, Buehler et al. (2007) proposed a formal scientific requirement for a passive submillimeter-wave cloud ice mission. The requirements are that the low IWP should be less than 10 g/m², the high IWP should be less than 50%, and the particle diameter should be less than 50 μm. Li et al. (2016) investigated the effects of five ICS models from the single-scattering property database of (Hong et al., 2009) at frequencies ranging from 100 to 1000 GHz, namely aggregates, hollow columns, flat plates, rosettes and spheres (Van De Hulst, 1957), on the transmission characteristics of terahertz radiation. Results showed that the ice particle shape is one of the dominant factors affecting terahertz radiation. Lately, Fox et al. (2020) evaluated seven ice particle habits from the ARTS scattering database (Eriksson et al., 2018). They showed that the randomly-oriented large column aggregate performs best when simulating brightness temperatures between 183 and 664 GHz.

In summary, the physical method is based on the radiative transfer principle and relies on the forward physical model simulation and effective ice crystal scattering (ICS) model. Different assumptions of ice cloud microphysical properties (shape, size, orientation and particle size distribution of ice particles) in the forward physical model significantly affect the retrieval of the IWP and particle size of ice clouds. Therefore, a practical and representative ICS model is essential for the ice cloud remote sensing in the terahertz region.

Many aircraft observations have demonstrated that ice clouds comprise a complex and diverse range of non-spherical ice particles (Lawson et al., 2006, 2019; Liou, 1992). It is still challenging for one specific light-scattering method to precisely calculate the single-scattering properties for non-spherical particles with different size parameters (SZPs). The SZP is the ratio of the equivalent-volume sphere's circumference dimension (or π times particle maximum diameters) to the incident wavelength (Baran, 2012; Nakajima et al., 2009). So far, the light-scattering methods can be roughly divided into the Approximation Method (AM) and the Numerical simulation Method (NM). The AM method is based on ray-tracing techniques, including the Geometrical Optics Method (GOM) (Macke et al., 1996a; Takano and Liou, 1989), which is suitable for large particles. The NM includes the Finite Difference Time Domain (FDTD) (Yang and Liou, 1996b; Yee, 1966) and the DDA (Draine and Flatau, 1994; Yurkin and Hoekstra, 2007) methods, which are appropriate for small particles. Combined with the advantages of the AM and NM methods, several improved Geometrical Optics Approximation (GOA) methods, including the Geometric Optics Integral Equation (GOIE) method (Yang and Liou, 1996a), have been developed. Moreover, the boundary element method (Groth et al., 2015; Kleanthous et al., 2022) has been recently applied to complex ice particles. Based on the above-mentioned light-scattering calculation methods, a series of regular-shaped ICS models have been designed, including hexagonal columns, hexagonal plates, bullet rosettes, droxtals and so on (Yang et al., 2000b, 2013; Fu et al., 1999; Takano and Liou, 1989). Since the regular-shaped ICS models are not fully representative of the scattering properties of natural ice particles, researchers have developed complex ICS models. Yang et al. (2013) developed the surface-roughened non-spherical ICS models and applied them to the production of MODIS C6 ice cloud products (Platnick et al., 2017). C.-Labonnote et al. (2000, 2001) and Doutriaux-Boucher et al. (2000) developed an ICS database for the Inhomogeneous Hexagonal Monocrystal (IHM) containing embedded inclusions (air bubbles and aerosols). The IHM

database was applied for the ice cloud retrievals from the French satellite Polarization and Directionality of the Earth's Reflectance (POLDER) measurements (Deschamps et al., 1994). Furthermore, Baran and Labonnote (2007) and Baran et al. (2014a) developed an ensemble ice particle model made of hexagonal column ice particles for use in the Met Office Unified Model Global Atmosphere 5.0 (GA5) configuration (Baran et al., 2014b). Unlike the above-mentioned regular-shaped ICS

models, Letu et al. (2012) and Ishimoto et al. (2013) analyzed ice particles of different shapes and sizes from many NASA aircraft observations and developed an irregular-shaped complex Voronoi ICS model. The single-scattering property database of the Voronoi ICS model in the visible and infrared regions using a combined FDTD, GOIE and GOM approach. The Voronoi ICS model has been adopted for generating official ice cloud products for the Second Generation gLobal Imager (SGLI)/Global Change Observation Mission-Climate (GCOM-C) (Letu et al., 2012, 2016; Nakajima et al., 2019),

AHI/Himawari-8 (Letu et al., 2018) and the Multi-Spectral Imager (MSI)/Earth Cloud Aerosol and Radiation Explorer (EarthCARE) satellite programs (Illingworth et al., 2015), which will be launched in 2023. Furthermore, Li et al. (2022) showed the effectiveness of the Voronoi ICS model in describing ice clouds' global visible and infrared radiative properties in the Community Integrated Earth System Model (CIESM). As a result, researchers have proven the effectiveness of the database of the Voronoi ICS model in the visible and infrared regions in the ice cloud satellite remote sensing and climate

model applications (Letu et al., 2016). Recently, Letu et al. (2012) and Ishimoto et al. (2013) have successfully extended the spectral range of the single-scattering property database of the Voronoi ICS model to the terahertz wave region. The database of the Voronoi ICS model in the terahertz region was adopted by Baran et al. (2018) as standard data for the modelling and evaluation of the ISMAR (International Sub-Millimeter wave Airborne Radiometer), which the European Space Agency (ESA) and Met Office jointly developed (Kangas et al., 2014). The results showed good evaluation

performance of the Voronoi ICS model. However, the effectiveness of the Voronoi ICS model in the terahertz region in a practical retrieval of ice cloud microphysical properties from terahertz radiation has yet to be discovered.

Motivated by the abovementioned situations, this study aims to apply the Voronoi ICS model to the ice cloud remote sensing retrieval of IWP and particle size from aircraft-based terahertz radiation measurements. To achieve this goal, we use the Voronoi ICS model to create a parameterization scheme (referred to as the Voronoi ICS scheme hereafter) for the ice cloud

scattering property in the terahertz region. The Voronoi ICS scheme is employed in the RSTAR radiative transfer model (Nakajima and Tanaka, 1986, 1988) to build the LUT for upward terahertz brightness temperature differences between the cloudy and clear sky (BTDs) at the top of the ice cloud (TOC) between multi-channel frequencies. The LUT based on the Voronoi ICS model is compared with that of the Sphere and Hexagonal Column ICS models. Finally, the CoSSIR-MCBI results evaluates the retrieval results searched iteratively from the LUTs for the Voronoi, Sphere and Hexagonal Column

models. This paper is organised as follows: Section 2 introduces the data and radiative transfer model used in this study. Section 3 describes the ice cloud parameterization scheme, retrieval algorithm and validation with the CoSSIR-MCBI algorithml. Section 4 presents the results of the retrieved IWP and particle size, a comparison of the Voronoi, Sphere and Hexagonal Column ICS models and the validation of the retrieval algorithm. Section 5 presents the conclusions of this study.

## 2 Data and model

### 2.1 Single-scattering property database for the Voronoi, Sphere and Column ICS models

This study used the single-scattering property database of three ice crystal habits (Voronoi, Sphere and Hexagonal Column) in the terahertz radiative transfer forward simulation. For the Voronoi ICS model, the single-scattering property database contains 31 ice particle sizes ranging from 0.25 to 9300 μm and covers 20 terahertz channels with frequencies ranging from 10 to 874 GHz (see Table 1), corresponding to wavelengths from 0.03 to 3cm. The Hexagonal Column ICS model (referred to as the Column ICS model hereafter) is randomly-oriented and was defined by Yang et al. (2000a). For the Column ICS model, the aspect ratios $a/L$ (defined as the ratio of the semiwidth $a$ of a particle to its length $L$) are defined as 0.35 and 3.48, respectively, when $L$ is less than 100 μm and greater than or equal to 100 μm. The single-scattering property database of the Column ICS model used in the study was developed by Hong (2007, 2009) using the DDA method. For the Column ICS model, the single-scattering property database includes 38 maximum particle dimensions ranging from 2 to 2000 μm and covers 21 terahertz channels with frequencies ranging from 90 to 874 GHz (see Table 1).

The single-scattering properties, including the extinction efficiency, single-scattering albedo and asymmetry factor of the Voronoi, Sphere and Column ICS models in the terahertz region, are used to calculate the terahertz scattering properties of ice clouds. For the Voronoi ICS model, the single-scattering properties are derived from the single-scattering property database developed by Letu et al. (2012, 2016) and Ishimoto et al. (2012) using a combination of FDTD and DDA methods. The DDA method is used to calculate the single-scattering properties of moderate ice particles (SZP>30). The FDTD method is used for small ice particles (SZP<30). As the particle size increases, the shape of the Voronoi ICS model changes and becomes complicated. The geometrical characteristics of Voronoi ICS model shape with increasing ice crystal size have been shown in Fig. 3 and Table 1 in Ishimoto et al. (2012). The mass-dimension and area-dimension power-law relationships of the Voronoi ICS model are defined by Ishimoto et al. (2012) and are described in Eq. (1)-(3) as shown below.

$$m = 0.00528D^{2.1} \textit{ (in cgs)} \tag{1}$$

$$A_r = 4G/\pi D^2 \tag{2}$$

$$A_r = 0.20D^{-0.29} \tag{3}$$

where $m$ is the mass, $G$ is the cross-sectional area, $A_r$ is the area ratio and $D$ is the maximum particle dimension of the Voronoi ICS model. The single-scattering property database of the Sphere ICS model is developed based on the exact solution of the Lorentz-Mie theory (Van De Hulst, 1957). The Sphere ICS database contains the same ice particle sizes and terahertz frequencies as the Voronoi ICS database. For the Voronoi and Sphere ICS models, calculations of their single-scattering properties utilized the real and imaginary parts of ice from the newest library of the refractive index provided by Warren and Brandt (2008). The refractive indices of the Voronoi and Sphere ICS models at frequencies of 10-874 GHz are computed at a temperature of 266 K, according to Warren and Brandt (2008). The refractive indices of the Column ICS model at frequencies of 90-340 GHz are derived from Warren (1984) at a temperature of -30 ℃.

## 2.2 Airborne measurements and auxiliary data

The measured terahertz brightness temperature from CoSSIR/ER-2 aircraft during the TC4 mission on 17 and 19 July 2007 are utilized in this study (available https://espoarchive.nasa.gov/archive/browse/tc4/ER2). During the TC4 field campaign, the CoSSIR measured brightness temperatures in channels from 183.3 to 873.6 GHz (183.3 ± 1.0, 3.0, 6.6, 220, 380.2 ± 1.8, 3.3, 6.2, 640 V, and 874 GHz), all with matched beamwidths about 4° (Evans et al., 2012). According to the studies by Li et al. (2016) and Liu et al. (2021), the atmospheric windows are near 640 and 874 GHz, and the atmospheric absorption peaks are near 380 GHz. The leading absorbing gases are water vapour and ozone. Therefore, both the 640 and 874 GHz brightness temperature are affected by ice clouds, while the brightness temperature of 380 GHz is insensitive to ice cloud microphysical properties. Hence, the 380 minus 640 GHz brightness temperature differences can highlight the brightness temperature depression caused by ice clouds. And the 640 minus 874 GHz brightness temperature differences can reflect the difference in the scattering properties of differently shaped ice clouds. According to Li et al. (2016), the differences between 640 and 874 GHz also can offset the regional errors due to different latitudes and atmospheric profiles. In this study, we choose the centre frequencies of 380, 640 and 874 GHz.

The particle size distributions (PSDs) describe the relationship between ice particle size and particle number concentration and are essential in the calculation of the average scattering properties of ice clouds. In this study, we select 14,408 groups of PSDs from aircraft observation sampling data (available at http://stc-se.com/data/bbaum/Ice_Models/microphysical_data.html) (Heymsfield et al., 2013b), which are based on 11 field flight observation experiments covering ice cloud observations in mid-latitude and tropical regions. The 11 field programs span a wide range of locations (ranging from 12°S to 70°N latitudes and from 148°W to 130°E longitudes) and encompass the temperature range 0° to -86°C, with altitudes from near the surface to 18.7 km. This dataset is representative of the wide range of conditions where ice clouds are found in the troposphere and lower stratosphere on a near-global scale (Li et al., 2022; Heymsfield et al., 2013a). For the fitting of PSDs for the Voronoi, Sphere and Column ICS models, we adopted the gamma distribution following Heymsfield et al. (2013a) in the form of Eq. (4):

$$n(L) = N_0 L^\mu e^{-\lambda L} , \qquad\qquad\qquad (4)$$

where $L$ is the maximum particle dimension, $n(L)$ is the particle concentration per unit volume (e.g., 1/cm³), $N_0$ is the intercept, $\lambda$ is the slope, and $\mu$ is the dispersion. The physical meaning of the PSDs is that $n(L)$ times $dL$ is the number of particles per unit area.

The total water vapour and ozone column data provided by the ERA5 reanalysis data are used as input data for the radiative transfer model to simulate clear-sky brightness temperature. The ERA5 reanalysis data is the fifth generation reanalysis data product developed by the European Centre for Medium-Range Weather Forecasts (ECMWF) (Hans et al., 2019). The total water vapour and ozone column data used have a horizontal resolution of 0.25° × 0.25° and a temporal resolution of 1 hour. The retrieved results are validated against the IWP and $D_{me}$ (Evans et al., 2005) from the CoSSIR-MCBI algorithm during the same period. The IWP and $D_{me}$ are available at https://espoarchive.nasa.gov/archive/browse/tc4/ER2.

## 2.3 Radiative transfer model

The RSTAR radiative transfer model is a set of numerical radiative transfer models developed by Nakajima and Tanaka (1986, 1988) for the plane-parallel atmosphere. The calculated wavelengths can cover from 0.17 to 1000 μm, and the assumed plane-parallel atmosphere could be divided into 50 layers from sea level to a maximum altitude of 120 km. The RSTAR model contains six atmospheric profiles (tropical, mid-latitude summer, mid-latitude winter, high-latitude summer, high-latitude winter, and U.S. standard atmosphere), including vertical profiles of temperature, pressure, water vapour, and ozone. Calculating gas absorption in RSTAR is based on the K-distribution method developed by Sekiguchi and Nakajima (2008), which considers important atmospheric radiative gases ($H_2O$, $CO_2$, $O_3$, $N_2O$, CO $CH_4$, etc.) and trace gases. The K-distribution method parameters were obtained from the HITRAN-2004 database. The RSTAR radiative transfer model assumes the simulated scene is composed of a homogeneous ice cloud layer.

## 3 Method

Figure 1 shows the general flowchart for the inversion of the IWP and $D_{me}$ of ice clouds using the CoSSIR brightness temperature measurements. For convenience, the difference between the 640 GHz BTD and the 874 GHz BTD is simplified to $BTD_{2-3}$. And the difference between the 380 GHz BTD and the 640 GHz BTD is simplified to $BTD_{1-2}$. We named the difference between the $BTD_{1-2}$ and $BTD_{2-3}$ as $BTD_{1-3}$. Firstly, based on the single-scattering property database of the Voronoi, Sphere and Column ICS models, the atmospheric radiative transfer model RSTAR is used to build LUTs of top of cloud (TOC) $BTD_{2-3}$ and $BTD_{1-3}$ for three ICS models, respectively. With the assumptions of a priori value of IWP and $D_{me}$, the initial TOC $BTD_{2-3}$ and $BTD_{1-3}$ simulations can be searched from the LUTs for three ICS models. Secondly, we use the RSTAR to construct a clear-sky LUT of TOC brightness temperature (BT) for 380, 640 and 874 GHz, respectively. The inputs are different ozone and water vapour column values under clear sky conditions (see section 3.2 for details). Then, the ERA5 reanalysis data is used to estimate the clear-sky TOC BT based on the clear-sky LUT. The measured cloudy-sky TOC BT of 380, 640 and 874 GHz are obtained from the CoSSIR measurements. Consequently, the TOC $BTD_{2-3}$ and $BTD_{1-3}$ measurements can be calculated using the measured cloudy-sky TOC BT minus the interpolated clear-sky TOC BT. Finally, the cost function will provide an estimate of the coherence between the measured and simulated TOC $BTD_{2-3}$ and $BTD_{1-3}$. The IWP and $D_{me}$ are obtained by using Gaussian Newton non-linear optimization estimation method. The retrieved results are validated against the IWP and $D_{me}$ retrieved from the CoSSIR-Evans algorithm. In the RSTAR forward simulation, the input parameter of the ice particle size is $R_e$, which is defined by Eq. (5):

$$R_e = \frac{\int_0^\infty r^3 n(r)\,dr}{\int_0^\infty r^2 n(r)\,dr},$$  (5)

where $r$ is the equivalent-volume sphere's particle radius, $n(r)$ is the particle size distribution. According to Sieron et al. (2017), a mass-weighted size $D_{me}$ would be a more appropriate parameter size than the area-weighted size for describing the PSDs. The $D_{me}$ is given by Eq. (6):

$$D_{me} = \frac{\int_0^\infty Dm(D)n(D)dD}{\int_0^\infty m(D)\,n(D)dD}, \tag{6}$$

where $D$ is the maximum particle dimension of ice particles, $m$ is the particle mass, $n(D)$ is the particle size distribution. To validate the retrieved $D_{me}$ using the $D_{me}$ from the CoSSIR-Evans algorithm, the transformation from the $R_e$ to the $D_{me}$ is necessary due to the different definitions of $R_e$ and $D_{me}$. Hence, we developed a conversion relationship between the $R_e$ and $D_{me}$ combined with different particle size distributions. Based on the integration of $r$ and $D$ over 14,408 PSDs, multiple groups of $R_e$ and $D_{me}$ are calculated according to Eq.(5) and (6). Numerical fitting was used to build a relationship between the $R_e$ (independent variable) and $D_{me}$ (dependent variable) over 14,408 PSDs. The conversion formulae between $R_e$ and $D_{me}$ is demonstrated as follows,

$$D_{me} = a + bR_e + cR_e^2, \tag{7}$$

where $a$, $b$ and $c$ are fitting coefficients obtained by numerical fitting and provided as input. Based on this relationship, we have unified all the $R_e$ into $D_{me}$ for comparability for the Voronoi, Sphere and Column ICS models.

**3.1 Parameterization of ice cloud optical properties**

To apply the Voronoi, Sphere and Column ICS models to the RTSAR radiative transfer model for forwarding simulation, three parameterization schemes (referred to as the Voronoi, Sphere and Column ICS schemes hereafter) for the scattering properties of ice clouds in the terahertz spectrum need to be constructed. The ice cloud optical properties depend on the single-scattering properties of the ICS and the PSDs. The effective ice cloud particle size measures the average ice cloud particle size for a given PSD. Different methods have been used to determine the effective ice cloud size, and according to Baum et al. (2005a, 2005b), the effective particle diameter for a given PSD is determined as Eq. (8):

$$D_e = \frac{3}{2} \frac{\int_{L_{min}}^{L_{max}} V(L)n(L)dL}{\int_{L_{max}}^{L_{max}} A(L)n(L)dL}, \tag{8}$$

where $D_e$ is the effective particle diameter, $V$ and $A$ are the volume and projected area of Voronoi and Sphere models. Then, the spectral ice cloud optical properties (mass-averaged extinction coefficients, single-scattering albedo, asymmetry factor and mass-averaged absorption coefficients) for the Voronoi and Sphere ICS schemes are calculated for all PSDs given by Eq. (9)-(12):

$$K_{ext}(\lambda) = \frac{\int_{L_{min}}^{L_{max}} Q_{ext}(\lambda,L)A(L)n(L)dL}{\rho_{ice} \int_{L_{min}}^{L_{max}} V(L)n(L)dL}, \tag{9}$$

$$\varpi(\lambda) = \frac{\int_{L_{min}}^{L_{max}} Q_{sca}(\lambda,L)A(L)n(L)dL}{\int_{L_{min}}^{L_{max}} Q_{ext}(\lambda,L)A(L)n(L)dL}, \tag{10}$$

$$g(\lambda) = \frac{\int_{L_{min}}^{L_{max}} g(\lambda,L)\sigma_{sca}(\lambda,L)n(L)dL}{\int_{L_{min}}^{L_{max}} \sigma_{sca}(\lambda,L)n(L)dL}, \tag{11}$$

$$K_{abs}(\lambda) = \frac{\int_{L_{min}}^{L_{max}} Q_{abs}(\lambda,L)A(L)n(L)dL}{\rho_{ice} \int_{L_{min}}^{L_{max}} V(L)n(L)dL}, \tag{12}$$

where $K_{ext}(\lambda)$ are spectral mass-averaged extinction coefficients (m$^2$/g), $\varpi(\lambda)$ is spectral single-scattering albedo, $g(\lambda)$ is spectral asymmetry factor and $K_{abs}(\lambda)$ are spectral mass-averaged absorption coefficients (m$^2$/g). $Q_{ext}$, $g$, $Q_{sca}$ and $Q_{abs}$ are extinction efficiency, asymmetry factor, scattering efficiency and absorption efficiency for Voronoi, Sphere and Column models. Based on the parameterized scattering properties of ice clouds and the effective particle diameter of ice clouds, we developed a parameterization scheme for the scattering properties of ice clouds by establishing the spectral bulk scattering properties of ice clouds as functions of the effective particle diameter of ice clouds. The least squares method is used to obtain first-order and third-order polynomial fitting equations according to Eq. (13)-(16):

$$K_{ext}(\lambda) = a_0 + a_1/D_e , \tag{13}$$

$$\varpi(\lambda) = b_0 + b_1 D_e + b_2 D_e^2 + b_3 D_e^3 , \tag{14}$$

$$g(\lambda) = c_0 + c_1 D_e + c_2 D_e^2 + c_3 D_e^3 , \tag{15}$$

$$K_{abs}(\lambda) = d_0 + d_1 D_e + d_2 D_e^2 + d_3 D_e^3 , \tag{16}$$

where $a_i$, $b_j$, $c_j$ and $d_j$ ($i$=0, 1; $j$=0, 1, 2, 3 ) are fitting coefficients and are spectral functions of the terahertz wavelength. Values of the above coefficients for the Voronoi ICS scheme are listed in appendix A (Tables A.1, A.2, A.3, and A.4).

## 3.2 Look-up table

Before constructing the LUT, the sensitivity analysis of different terahertz brightness temperatures to the IWP and $D_{me}$ of ice clouds needs to be analyzed to build a representative and efficient LUT. According to the sensitivity results (see section 4.3), the BTDs between the cloudy and clear-sky conditions at 380, 640 and 874 GHz frequencies are simplified to BTD$_1$, BTD$_2$ and BTD$_3$, respectively. And the differences between the two BTDs are represented by a dash. Based on the Voronoi, Sphere and Column ICS schemes, the RSTAR is used to construct three LUTs (Voronoi, Sphere and Column LUTs) of BTD$_{2-3}$ and BTD$_{1-3}$ for cloudy and clear-sky conditions. For the cloudy-sky LUT, the number of the IWP is set to 12, and the range of values is defined in log10 space from 0 to 3.36 in steps of 0.28. The number of $R_e$ is set to 6, and the range of values is defined in log10 space from 1.6 to 3.1 in steps of 0.25. For the clear-sky LUT, the number of total ozone columns is set to 7, and the range of values is from 200 to 500 in steps of 50. To be consistent with aircraft observations of terahertz brightness temperatures, BTD$_{2-3}$ and BTD$_{1-3}$ are simulated at the mean altitude (20 km). The U.S. standard atmospheric profile is used in the simulation, and the cloud top temperature is assumed to be the same as the atmospheric temperature at that level. The surface is assumed to be a black body (emissivity equals 1) with a temperature of 288.15 K.

## 3.3 Optimal estimation inversion method

Based on the terahertz BTDs LUTs for the Voronoi, Sphere and Column ICS schemes established in the previous section, the IWP and $D_{me}$ are retrieved using an optimization method with Gaussian Newtonian nonlinear iterations. Based on the

300 atmospheric radiative transfer transmission in the terahertz spectrum, if the background field under clear-sky conditions is subtracted from the cloudy-sky condition, the TOC BTDs are only functions of the cloud microphysical property parameters, which are given by Eq. (17)-(18):

$$Y = F(X) + \epsilon ,$$ (17)

$$X = \begin{pmatrix} IWP, \\ D_{me} \end{pmatrix}, Y = \begin{pmatrix} BTD_{2-3}, \\ BTD_{1-3} \end{pmatrix},$$ (18)

where $X$ is the vector-matrix composed of the variables of the IWP and $D_{me}$ to be solved. $Y$ is the vector composed of the two BTDs and the uncertainty vector. The vector $\epsilon$ represents the uncertainties that are attached to the measurements (i.e.

instrumental accuracy) and to the radiative transfer forward model (i.e.approximation errors in the radiative transfer model). Following Marks and Rodgers (1993), a good convergence can be obtained when the value of the cost function is lower than the size of the measurement vector. Since there is no robust a priori for IWP and $D_{me}$, we selected an average value as the initial value for a priori value of the IWP and $D_{me}$. Assuming that $Y$ is the value of two BTDs measured by the aircraft, the inverse $X$ is the minimum value for solving Eq. (19):

$$J(X) = min\{\|y - F(X)\| + \gamma \|X - X_a\|\} ,$$ (19)

where $X_a$ is a vector consisting of the prior estimates of the IWP and $R_e$, $\gamma$ is a Lagrangian operator, $min$ denotes the minimal value of solving this function, and the value of $X$ at the minimal value is considered as the retrieved result. For the solution method of the nonlinear problem in the inversion process, the Newton nonlinear iterative method is usually used. It has a faster inversion speed and higher solution accuracy, and its iterative form follows Eq. (20):

$$X_{i+1} = X_i - J'(X_i)^{-1} J'(X_i) ,$$ (20)

where the $i$ represents the number of iterations, and the superscripts denote the first-order and second-order derivations, then

the iterations start from the a priori initial estimate until the convergence criterion is satisfied or when the number of iterations satisfies the required number of iterations, then the iteration is stopped, and the solution results are obtained. The final inversion results are validated using the results from the CoSSIR-MCBI algorithm.

## 4 Results

### 4.1 Single-scattering property database for the Voronoi, Sphere and Column ICS models

Figure 2 compares the extinction efficiency, single-scattering albedo and asymmetry factor, varying with the SZP for the Voronoi, Column and Sphere ICS models at 325 and 874 GHz. For small ice particles with SZPs less than 0.1, the single-scattering properties are small and barely influenced by the shape of the ice particles. This is because the single-scattering properties of ice particles are close to Rayleigh scattering when the SZP is small. As particle size increases, particle scattering is predominantly Mie scattering, and the sensitivity of the single-scattering properties to the ice crystal habits

becomes pronounced so that the ice crystal shape contributes to the large differences for large particle sizes. The single-scattering properties of the Voronoi and Column models vary more smoothly than those of the Sphere model. As shown in

Figure 2, the smooth curve of the Voronoi model reflects that FDTD and DDA methods provide good continuity for the transition between different size parameters of the Voronoi model. For large particles (SZP > 3) at 874 GHz, the Voronoi model has the highest extinction efficiency than the other two models. At 325 GHz, the single-scattering albedo of the Voronoi model is close to the Sphere model and is higher than the Column model. At 874 GHz, the single-scattering albedo of the Voronoi model is close to the Column model and is higher than the Sphere model. The Voronoi model has the highest asymmetry factor than the other two models at 325 and 874 GHz. On the one hand, the higher extinction efficiency and single-scattering albedo of the Voronoi ICS model for large particles are possibly due to the multifaceted shapes of the Voronoi ICS model, which can result in significant side and backward scattering and increase the scattered energy. On the other hand, the higher asymmetry factor of the Voronoi ICS model for large particles is possible because the scattered energy is dominated by diffraction. The diffracted energy is concentrated in the forward direction, leading to a large asymmetry factor of the Voronoi ICS model.

Figure 3 shows the contours of the single-scattering properties for the Voronoi, Sphere and Column ICS models over 20 terahertz wavelengths (0.03-0.3 cm) and 31 particle sizes (1-270 μm). The sharp changes in the single-scattering albedo for the Voronoi, Sphere and Column ICS models can be seen from small to large ice particles. For small particles in the low-frequency channels (long wavelengths), their scattering is mainly Rayleigh scattering with significant absorption effects. The absorption energy is proportional to the volume of ice particles and is barely affected by the ice particle shape. In the high-frequency channels (short wavelengths), the Mie-scattering plays a leading role, and the scattering function plays a dominant role. Especially for large ice particles, the single-scattering albedo is close to one, and the influence of the ice particle shape becomes obvious. Figure 4 shows the differences in extinction efficiency, single-scattering albedo and asymmetry factor between the Voronoi and the other two ICS models as functions of the effective radius and terahertz wavelength. Overall, the extinction efficiency of the Voronoi model is higher than the other two ICS models. The asymmetry factor of the Voronoi model is higher than the other two ICS models in the low-frequency channel for large ice particles.

Figure 5 displays the scattering phase functions of the Voronoi, Sphere and Column ICS models. The variation of the scattering phase function for the Voronoi model tends to be more dramatic compared to the Sphere and Column ICS models. The scattering phase function is axisymmetric about the 90° scattering angle for small ice particles, which can be approximated as Rayleigh scattering. For small ice particles, the scattering phase function shows extreme values in the forward (0°) and backward (180°) directions and shows minimal values in both side directions (90° and 270°). As the ice particle size increases, the forwarding scattering is significantly larger than the side and backward scattering and gradually tends to be more Mie-scattering. For the Sphere model in the high-frequency channels, the forward scattering increases with the increase in particle size. In the low-frequency channels, the forward scattering remains almost constant with the increase in particle size. Compared to the Voronoi and Sphere ICS models, the Column ICS model shows smaller phase functions. For the Voronoi model, the low values of the scattering phase function move towards large scattering angles with the increase in particle size.

## 4.2 Bulk scattering properties of ice clouds in the terahertz region

Based on the single-scattering properties of the Voronoi, Sphere and Column ICS models, the Voronoi, Sphere and Column ICS schemes are developed in the terahertz region with the integration over both PSDs and terahertz frequencies. The calculated bulk scattering properties of ice clouds include the mass extinction coefficients, single-scattering albedo, asymmetry factor and mass absorption coefficients. Figure 6 compares the calculated bulk scattering properties of the Voronoi, Sphere and Column ICS schemes over 14,408 PSDs at four terahertz frequencies (325, 448, 664 and 874 GHz). The bulk scattering properties of ice clouds depend on the effective diameter and the terahertz frequency. The single-scattering albedo increases linearly with the effective diameter for the Sphere ICS scheme. For the Voronoi ICS scheme, the single-scattering albedo increases for the effective diameter smaller than 100 μm and approaches 0.95 with effective diameters exceeding 100 μm. The Column ICS scheme has the lowest single-scattering albedo compared to the other two schemes for small effective diameters and gradually exceeds the other two schemes with the increase of the frequency and effective diameter. The mass extinction coefficients obtained from the three schemes show a uniformly positive correlation with the effective diameter and the terahertz frequency. At four terahertz frequencies, the mass extinction coefficients of the Voronoi ICS scheme are close to the Column ICS scheme for all effective diameters. The Sphere ICS scheme has the highest mass extinction coefficients compared to the other two schemes for all effective diameters and four terahertz frequencies. The Voronoi ICS scheme has a similar asymmetry factor to the Sphere ICS scheme at low frequency and becomes the highest compared to the other two schemes at high frequency. The Column ICS scheme has the lowest asymmetry factor compared to the other two schemes for all effective diameters. For all effective diameters, the mass absorption coefficients of the Sphere ICS scheme have the strongest absorption effect than the other two schemes and decrease with increasing effective diameter. The Voronoi ICS scheme has the lowest mass absorption coefficients compared to the other two schemes for effective diameter smaller than 100 μm and becomes higher than the Column ICS scheme with increasing effective diameters. Overall, the bulk scattering properties for all schemes increase with increasing frequencies. This is mainly because the scattering properties of ice clouds are more significant as the wavelength decreases.

## 4.3 Sensitivity results

We discuss the sensitivity of the TOC $BTD_{2-3}$ and $BTD_{1-3}$ to the IWP and $D_{me}$ based on the Voronoi, Sphere and Column ICS schemes, respectively, in the RSTAR model. Figure 7 shows that the $BTD_{2-3}$ and $BTD_{1-3}$ are positively correlated with the IWP and $D_{me}$. For the Voronoi and Column ICS models, the $BTD_{2-3}$ and $BTD_{1-3}$ show monotonically increasing relationships with the increase in IWP. For ice clouds with the $D_{me}$ larger than 200 μm, the 50 to 1000 μm particle sizes can lead to approximately 0-20 K $BTD_{2-3}$ and 0-30 K $BTD_{2-3}$. The $BTD_{2-3}$ and $BTD_{1-3}$ increase with the increase of the $D_{me}$ and is close to a constant value for $D_{me}$ larger than 600 μm. As shown in Figure 8, the Voronoi ICS scheme has higher $BTD_{2-3}$ and $BTD_{1-3}$ compared to the Sphere and Column ICS scheme. In summary, the TOC $BTD_{2-3}$ and $BTD_{1-3}$ have a strong sensitivity to the

IWP and $D_{me}$, especially for moderate to large ice particles and large IWP. The different ICS schemes vary considerably in their modeling of terahertz brightness temperatures.

Figure 9 exhibits the variation of the TOC BTDs at different IWPs and $D_{me}$ for 380, 640 and 874 GHz frequencies based on the RSTAR model. The $BTD_{1-3}$ has a strong sensitivity to the IWP and increases with the increase of the IWP, and the slope shows that the increase is pronounced when the IWP is less than 600 g/m$^2$. The $BTD_{2-3}$ has a strong sensitivity to the $D_{me}$ and increases with the increasing $D_{me}$. By comparing the LUTs of the Voronoi, Sphere and Column ICS models, it is found that at the same $BTD_{2-3}$, the Sphere LUT has the highest IWP compared to the Column and Voronoi LUT, while the Voronoi LUT has the lowest. The $D_{me}$ of the Sphere LUT is smaller than that of the Column and Voronoi LUT for the same $BTD_{1-3}$. This is mainly due to the higher single-scattering albedo and asymmetry factor of the Voronoi ICS model at low frequencies, resulting in stronger forward scattering energy and a larger BTD than the Sphere and Column ICS model for the same IWP and $D_{me}$.

### 4.4 Inversion and validation results

To develop the inversion method from the CoSSIR measurements of the terahertz brightness temperature, we need to perform a quantitative simulation to test the accuracy of the algorithm. Figure 10 shows the validation results of the inversion algorithm using 2000 groups of $BTD_{2-3}$ and $BTD_{1-3}$ simulated by RSTAR using 2000 randomly generated IWP and $D_{me}$. The results show that most of the validation results of the red density lie on the 1:1 line. The correlation coefficients of the IWP and $D_{me}$ are 0.94 and 0.99, and the mean absolute errors (MAEs) are 35.46 g/m$^2$ and 8.56 µm, respectively. The test results meet the formal scientific mission requirement of ice cloud remote sensing retrieval, according to Buehler et al. (2007). The high accuracy of the validation results proves the effectiveness and accuracy of the inversion algorithm.

We adopt the CoSSIR-MCBI results as the benchmark for evaluating the retrieved IWP and $D_{me}$ based on three schemes using the new inversion algorithm. To quantify the differences in the retrieval results between the three schemes and the CoSSIR-MCBI, we use the CoSSIR-MCBI results minus the retrieved results of the Voronoi, Column and Sphere ICS schemes, respectively, to analyze their differences. Figure 11 (a) and (b) show the CoSSIR-MCBI results on 19 July 2007 overlaid with the retrieval results of the IWP and $D_{me}$ based on the Voronoi scheme. Red and blue dots, respectively, represent the Voronoi ICS scheme and CoSSIR-MCBI results. Figure 11 (a) and (b) show that the overall performance shows good agreement for the Voronoi ICS scheme compared to the CoSSIR-MCBI results. The matching rates for the IWP and $D_{me}$ are 80.3% and 83.2%, respectively. Figure 11 (c) and (d) show the joint histogram of differences of the retrieved IWP and $D_{me}$ between the Voronoi, Sphere, Column schemes and the CoSSIR-MCBI results, respectively. Overall, the retrieved IWP and $D_{me}$ parameters from the Voronoi ICS scheme agree well with the CoSSIR-MCBI results for all cases.

Figure 12 shows the inversion results of IWP and $D_{me}$ based on the optimal estimation algorithm from the CoSSIR aircraft measured brightness temperature. For the Voronoi, Sphere and Column ICS models, the correlation coefficients of the retrieved IWP are 0.87, 0.67 and 0.74, with MAEs of 22.38 g/m$^2$, 23.55 and 22.91 g/m$^2$ and RMSE of 30.45 g/m$^2$, 35.22 and 32.64 g/m$^2$, respectively. The correlation coefficients of the $D_{me}$ are 0.83, 0.64 and 0.76, with MAEs of 18.46, 21.19 and

19.91 μm and RMSE of 24.57, 26.51 and 25.04 μm, respectively. Overall, the accuracy for the IWP and $D_{me}$ based on the Voronoi ICS model is better than the other two ICS models compared to the CoSSIR-MCBI results. The Sphere ICS model overestimates IWP and $D_{me}$ compared with the validation data. According to the sensitivity results of Figures 7 and 8, the Voronoi ICS scheme has higher $BTD_{2-3}$ and $BTD_{1-3}$ compared to the Sphere and Column ICS schemes, especially for large particles and IWP. This characteristic can be explicitly explained by the higher asymmetry factor of the Voronoi ICS model compared to the Sphere and Column ICS models. Thus, stronger forward scattering energy can be detected for the Voronoi ICS model than for the other two models. The look-up table of the Voronoi ICS model can cover more IWP and $D_{me}$. The brightness temperature variations of the Voronoi-shaped ice clouds are more prominent and sensitive to the IWP and $D_{me}$. Therefore, the Voronoi ICS model results are better than the other two models. According to the look-up table as shown in Figure 9, there are overlapping lines when $D_{me}$ is small ($D_{me} < 40$ μm) and large ($D_{me} > 140$ μm). When $BTD_{2-3}$ and $BTD_{1-3}$ data fall under such overlapping lines of the look-up table, this overlapping region can lead to obtaining the same IWP and $D_{me}$ when searching the look-up table.

## 5 Conclusions

In this study, we applied the irregular-shaped Voronoi ice crystal scattering (ICS) model to the ice cloud remote sensing retrieval of the ice water path (IWP) and particle size based on the Compact Scanning Sub-millimeter wave Imaging Radiometer (CoSSIR) terahertz radiation measurements. The bulk scattering property parameterization (Voronoi ICS scheme) in the terahertz region was developed based on the single-scattering properties of the Voronoi ICS database and 14,408 groups of particle size distributions from in-situ observations. The Voronoi ICS scheme was applied to the atmospheric radiative transfer model RSTAR and compared with the Sphere and Column ICS schemes to carry out the sensitivity analysis. We conducted the sensitivity analysis of brightness temperature differences between the cloudy and clear sky (BTDs) between 380, 640 and 874 GHz to the IWP and particle size. Based on the sensitivity analysis results, we built three terahertz multi-channel BTD look-up tables (LUTs) for the Voronoi, Sphere and Column ICS schemes using the RSTAR atmospheric radiative transfer model. Based on the three LUTs, we utilized the Gaussian Newton non-linear optimization estimation method to retrieve the IWP and particle size from the CoSSIR terahertz radiation measurements. Finally, the retrieval results were evaluated by the IWP and median mass diameter ($D_{me}$) derived from the CoSSIR-MCBI algorithm. The main conclusions were obtained as follows.

The bulk scattering properties of ice clouds in the terahertz region, including the mass extinction coefficients, single-scattering albedo, asymmetry factor and mass absorption coefficients of the Voronoi ICS scheme, were applied to the RSTAR model and were compared with the Sphere and Column ICS schemes. The results showed that the Voronoi ICS scheme has a distinct feature of lower absorption properties and higher asymmetry factors for larger particle sizes in the terahertz region. This feature could be related to the complex, multifaceted shape of the Voronoi ICS model and suggests

that the Voronoi ICS scheme can produce relatively stronger forward scattering and fewer absorption effects compared with the Sphere and Column ICS schemes.

The sensitivity analysis showed that the BTDs between 640 and 874 GHz are sensitive to the IWP variation, and the BTDs between 380, 640 and 874 GHz are sensitive to the particle size. The atmospheric absorption peak near 380 GHz and the atmospheric window near the 640 and 874 GHz can be effectively used for the IWP in the range of 50-200 $g/m^2$ and particle size of 50-300 μm.

A comparison of the results from the Voronoi, Sphere and Column ICS models shows that the results from the Voronoi ICS model are better than the Sphere and Column ICS models. The IWP and $D_{me}$ retrieved from the Voronoi ICS scheme showed higher consistency with CoSSIR-MCBI results than the other two schemes, with correlation coefficients of 0.87 and 0.83 for IWP and $D_{me}$, respectively. For the Voronoi ICS model, the mean absolute errors (MAEs) of the IWP and $D_{me}$ are improved by 5.0% and 12.8%, and RMSE is improved by 13.5% and 7.3%, respectively. The Sphere ICS scheme overestimates the IWP and $D_{me}$ by up to the MAEs of 23.55 $g/m^2$ and 21.19 μm, respectively. This is mainly due to the differences in the absorption efficiency and asymmetry factor in the single-scattering properties of ice particles, which have a significant impact on the description of the scattering and radiative properties of ice clouds in the terahertz region.

In conclusion, the analysis of terahertz BTDs between 380, 640 and 874 GHz exhibits obvious sensitivity to the IWP and particle size of ice clouds, which could complement visible/infrared and microwave spectra. The present work provides the potential utility of inferring the IWP and particle size of ice clouds using BTDs between 380, 640 and 874 GHz. With the LUT for BTDs between 380, 640 and 874 GHz, the retrieval of terahertz ice cloud properties from airborne measurements based on the irregularly-shaped Voronoi ICS models is newly developed. We find that the retrieval results based on the Voronoi ICS scheme present a better agreement with the CoSSIR-MCBI algorithm than the Sphere and Column ICS schemes. This study confirmed that the Voronoi ICS model has ice cloud inversion capabilities in the terahertz region, which may provide a reference for future use in aircraft-based and satellite-based terahertz ice cloud remote sensing applications.

**Data availability**

The CoSSIR/ER-2 aircraft data during the TC4 mission on 17 and 19 July 2007 are available at https://espoarchive.nasa.gov/archive/browse/tc4/ER2. The IWP and $D_{me}$ from the CoSSIR-MCBI algorithm are available at https://espoarchive.nasa.gov/archive/browse/tc4/ER2. The 14,408 groups of PSDs from 11 field flight observation experiments are available at http://stc-se.com/data/bbaum/Ice_Models/microphysical_data.html.

**Author contribution**

Ming Li developed the terahertz ice cloud remote sensing inversion algorithm for the IWP and particle size of ice clouds and evaluated the retrieval result by validating it against the results from the CoSSIR-MCBI algorithm. Ming Li is also

responsible for downloading auxiliary data and writing the initial draft of this manuscript. Husi Letu provided the single-scattering property database of Voronoi models in the terahertz region and assisted in developing the parameterization of ice cloud scattering properties in the RSTAR model. Husi Letu also designed the aims and structures of this study and guided the writing and revisions of the manuscript.

Hiroshi Ishimoto developed the single-scattering property database of Voronoi models in the terahertz region and helped in
the writing and revisions of the manuscript.

Takashi Y. Nakajima provided the atmospheric radiative transfer model RSTAR and is responsible for the optimization of the RSTAR model and assisted in the parameterization of ice cloud scattering properties.

Shulei Li and Lei Liu assisted in developing the ice cloud remote sensing retrieval algorithm of the IWP and particle size of ice clouds and provided the CoSSIR terahertz radiation measurement data, as well as helped with reviewing the manuscript.
Dabin Ji guided the development of the ice cloud remote sensing retrieval algorithm and helped with the review of the manuscript.

Huazhe Shang assisted in analyzing the results and guided the flowchart of the study, as well as reviewed the manuscript. Chong Shi assisted in designing the structures of this study, guided the writing of the paper and helped review the manuscript.

**Competing interests**

The authors declare that they have no conflict of interests.

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

Table Captions:

Table 1. The frequency channels and maximum particle dimensions of ice particles included in the single-scattering property database of the Voronoi and Column ICS models.


Table A.1. Coefficients in the fitting of terahertz mass extinction coefficients ($m^2$ /g).

Table A.2. Coefficients in the fitting of terahertz single-scattering albedo.

Table A.3. Coefficients in the fitting of terahertz asymmetry factor.

Table A.4. Coefficients in the fitting of terahertz mass-averaged absorption coefficients ($m^2$ /g).

Table 1. The frequency channels and maximum particle dimensions of ice particles included in the single-scattering property database of the Voronoi and Column ICS models.

| Voronoi ICS model | | Column ICS model | |
| --- | --- | --- | --- |
| Maximum particle dimension (µm) | Frequency (GHz) | Maximum particle dimension (µm) | Frequency (GHz) |
| 0.400E+00 | 0.10000E+02 | 0.200E+01 | 0.900E+02 |
| 0.100E+01 | 0.15000E+02 | 0.400E+01 | 0.118E+03 |
| 0.200E+01 | 0.18700E+02 | 0.600E+01 | 0.157E+03 |
| 0.300E+01 | 0.23800E+02 | 0.800E+01 | 0.166E+03 |
| 0.500E+01 | 0.31400E+02 | 0.100E+02 | 0.183E+03 |
| 0.750E+01 | 0.35000E+02 | 0.120E+02 | 0.190E+03 |
| 0.150E+02 | 0.50300E+02 | 0.150E+02 | 0.203E+03 |
| 0.250E+02 | 0.53750E+02 | 0.200E+02 | 0.220E+03 |
| 0.350E+02 | 0.55000E+02 | 0.250E+02 | 0.243E+03 |
| 0.450E+02 | 0.89000E+02 | 0.300E+02 | 0.325E+03 |
| 0.600E+02 | 0.94000E+02 | 0.400E+02 | 0.340E+03 |
| 0.700E+02 | 0.11875E+03 | 0.500E+02 | 0.380E+03 |
| 0.147E+03 | 0.16550E+03 | 0.600E+02 | 0.425E+03 |
| 0.225E+03 | 0.18331E+03 | 0.700E+02 | 0.448E+03 |
| 0.314E+03 | 0.22900E+03 | 0.800E+02 | 0.463E+03 |
| 0.419E+03 | 0.24300E+03 | 0.900E+02 | 0.487E+03 |
| 0.500E+03 | 0.32500E+03 | 0.100E+03 | 0.500E+03 |
| 0.623E+03 | 0.44800E+03 | 0.125E+03 | 0.640E+03 |
| 0.752E+03 | 0.66400E+03 | 0.150E+03 | 0.664E+03 |
| 0.867E+03 | 0.87400E+03 | 0.175E+03 | 0.683E+03 |
| 0.964E+03 | | 0.200E+03 | 0.874E+03 |
| 0.108E+04 | | 0.250E+03 | |
| 0.140E+04 | | 0.300E+03 | |
| 0.175E+04 | | 0.350E+03 | |
| 0.256E+04 | | 0.400E+03 | |
| 0.350E+04 | | 0.500E+03 | |

| | |
|---|---|
| 0.500E+04 | 0.600E+03 |
| 0.750E+04 | 0.700E+03 |
| 0.100E+05 | 0.800E+03 |
| 0.120E+05 | 0.900E+03 |
| 0.150E+05 | 0.100E+04 |
| | 0.110E+04 |
| | 0.120E+04 |
| | 0.130E+04 |
| | 0.140E+04 |
| | 0.160E+04 |
| | 0.180E+04 |
| | 0.200E+04 |

**Appendix A. Coefficients for Voronoi ICS scheme used in this study are tabulated in appendix**

Table A.1. Coefficients in the fitting of terahertz mass extinction coefficients ($m^2$ /g).

| Frequency (GHz) | $a_0$ ($m^2$/g) | $a_1$ ($m^3$/g) |
|---|---|---|
| 325 | 7.0891e-01 | -1.6965e+01 |
| 448 | 2.1347e+00 | -5.0405e+01 |
| 664 | 7.5009e+00 | -1.6770e+02 |
| 874 | 1.5790e+01 | -3.2850e+02 |

Table A.2. Coefficients in the fitting of terahertz single-scattering albedo.

| Frequency (GHz) | $b_0$ | $b_1$ | $b_2$ | $b_3$ |
| --- | --- | --- | --- | --- |
| 325 | -3.1317e-01 | 2.7448e-02 | -2.0449e-04 | 5.0815e-07 |
| 448 | -2.3947e-01 | 2.9461e-02 | -2.4145e-04 | 6.4366e-07 |
| 664 | -8.2857e-02 | 2.7985e-02 | -2.4357e-04 | 6.7691e-07 |
| 874 | 4.7425e-02 | 2.5164e-02 | -2.2395e-04 | 6.3152e-07 |

Table A.3. Coefficients in the fitting of terahertz asymmetry factor.

| Frequency (GHz) | $c_0$ | $c_1$ | $c_2$ | $c_3$ |
| --- | --- | --- | --- | --- |
| 325 | 2.2045e-02 | -8.2487e-04 | 2.5764e-05 | -4.7767e-08 |
| 448 | 1.0168e-02 | -5.1223e-05 | 3.0599e-05 | -8.0591e-08 |
| 664 | -4.4704e-02 | 3.5331e-03 | 1.2997e-05 | -7.2297e-08 |
| 874 | -1.1685e-01 | 8.8403e-03 | -3.0410e-05 | 2.6790e-08 |


Table A.4. Coefficients in the fitting of terahertz mass-averaged absorption coefficients ($m^2$/g).

| Frequency (GHz) | $d_0$ ($m^2$/g) | $d_1$ (m/g) | $d_2$ (1/g) | $d_3$ ($m^{-1}$/g) |
|---|---|---|---|---|
| 325 | 4.4262e-02 | 1.5585e-04 | 9.6647e-07 | -5.1271e-09 |
| 448 | 8.2110e-02 | 5.0544e-04 | 2.0336e-06 | -1.2945e-08 |
| 664 | 1.6909e-01 | 2.4299e-03 | 1.2784e-06 | -3.3930e-08 |
| 874 | 2.6509e-01 | 7.6295e-03 | -1.4488e-05 | -3.9275e-08 |


# Figure Captions:

Figure 1. The overall flowchart of the retrieval of the IWP and $D_{me}$ of ice clouds based on the Voronoi, Column and Sphere ICS models.

Figure 2. The extinction efficiency, single-scattering albedo and asymmetry factor as functions of the SZP for the Voronoi (solid blue line), Sphere (red dashed line) and Column (green dashed line) ICS models with a refractive index of $1.78 + 0.005i$ in the (a, c, e) 325 GHz and $1.78 + 0.015i$ in the (b, d, f) 874 GHz frequencies.

Figure 3. The comparison of the extinction efficiency, single-scattering albedo and asymmetry factor as functions of 31 particle sizes (1-270 μm) and 20 terahertz wavelengths (0.03-0.3 cm) for the (a, d, g) Voronoi, (b, e, h) Sphere and (c, f, i) Column ICS models.

Figure 4. The (a, c, e) Voronoi minus Sphere and the (b, d, f) Voronoi minus Column ICS model differences in (top row) extinction efficiency, (middle row) single-scattering albedo and (bottom row) asymmetry factor as functions of 31 particle sizes (1-270 μm) and 20 terahertz wavelengths (0.03-0.3 cm).

Figure 5. The scattering phase functions for ice particles with four sizes ($R_e$ = 30, 71, 107 and 153 μm) for the (a, d) Voronoi, (b, e) Sphere and (c, f) Column ICS models with a refractive index of $1.78 + 0.005i$ in the 325 GHz and $1.78 + 0.015i$ in the 874 GHz, respectively.

Figure 6. The comparison of the parameterized single-scattering albedo, mass extinction coefficients, asymmetry factor and mass absorption coefficients as functions of effective diameters for 325, 448, 664 and 874 GHz for the Sphere (blue line), Voronoi (orange line) and Column (green line) ICS models.

Figure 7. The $BTD_{2-3}$ and $BTD_{1-3}$ for the (a, d) Voronoi, (b, e) Sphere and (c, f) Column ICS models as functions of the IWP and $D_{me}$, respectively.

Figure 8. The $BTD_{2-3}$ and $BTD_{1-3}$ differences for the (a, c) Voronoi minus Sphere and (b, d) Voronoi
 minus Column ICS models as functions of the IWP and $D_{me}$, respectively.

Figure 9. The LUT of $BTD_{2-3}$ and $BTD_{1-3}$ for the (a) Voronoi, (b) Column and (c) Sphere ICS models varying with the logarithm of IWP and $D_{me}$. Grey dots in circles represent the randomly generated 2000 test data from the RSTAR model.

Figure 10. The scatterplots of the randomly generated 2000 test data and the retrieved results. The left panel and the right panel are for the IWP and $D_{me}$, respectively.

Figure 11. The scattered red dots are the retrieved (a) IWP and (b) $D_{me}$ (bottom row) validated by the
 results from the CoSSIR-MCBI algorithm (blue scattered dots). The joint histogram of differences of the retrieved (c) IWP and (d) $D_{me}$ between the Voronoi, Sphere, Column schemes and the CoSSIR-MCBI results, respectively.

Figure 12. The scatterplots of the retrieved IWP (top row) and $D_{me}$ (bottom row) against the CoSSIR-
 MCBI results for the Sphere (right column), Column (middle column) and Voronoi (left column) ICS models.

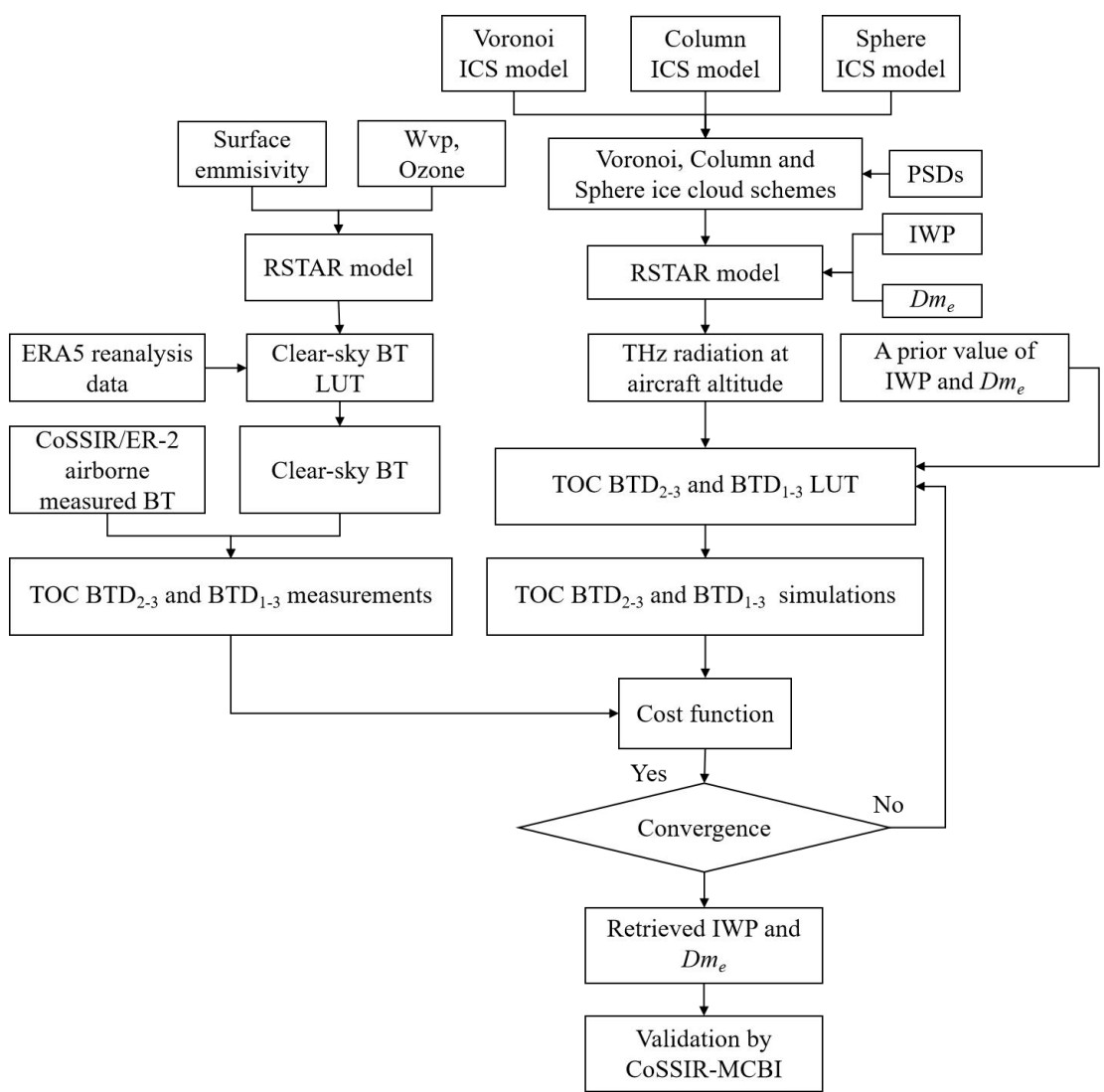


**Figure 1: The overall flowchart of the retrieval of the IWP and $D_{me}$ of ice clouds based on the Voronoi, Column and Sphere ICS models.**

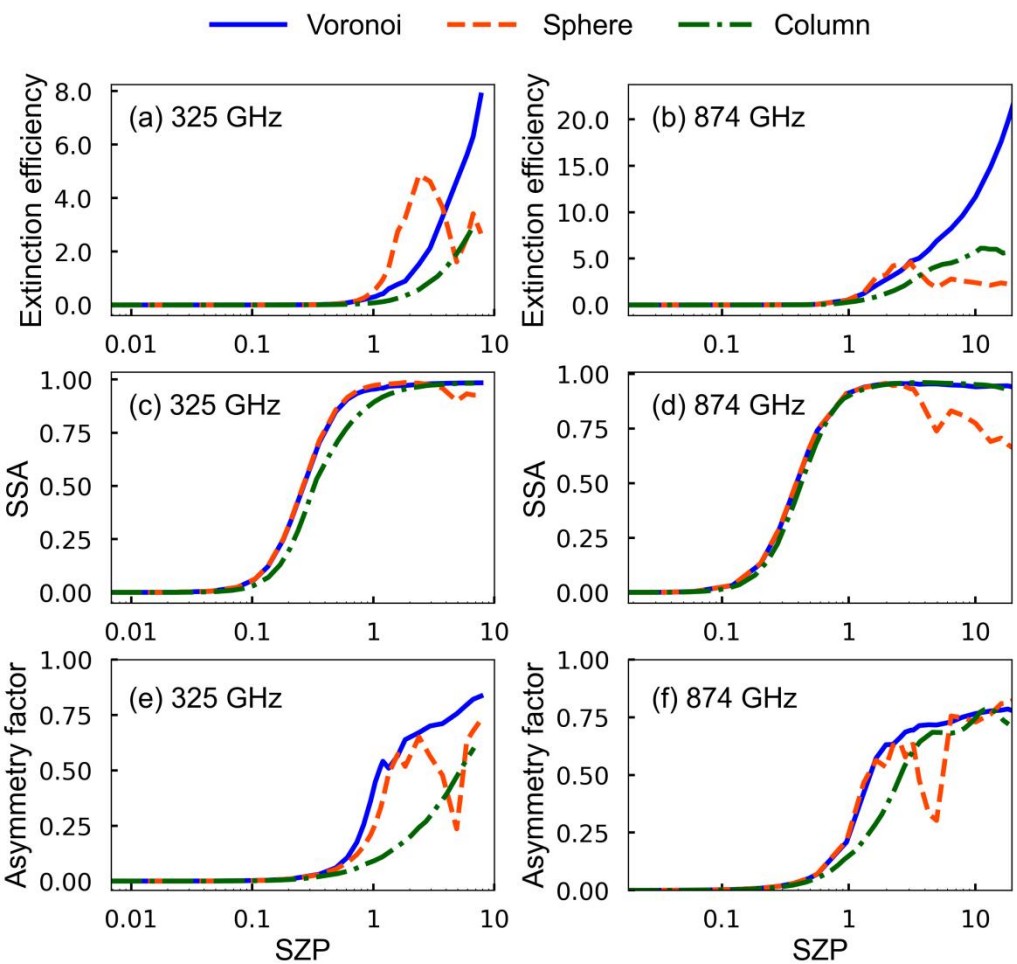

**Figure 2: The extinction efficiency, single-scattering albedo and asymmetry factor as functions of the SZP for the Voronoi (solid blue line), Sphere (red dashed line) and Column (green dashed line) ICS models with a refractive index of 1.78 + 0.005$i$ in the (a, c, e) 325 GHz and 1.78 + 0.015$i$ in the (b, d, f) 874 GHz frequencies.**

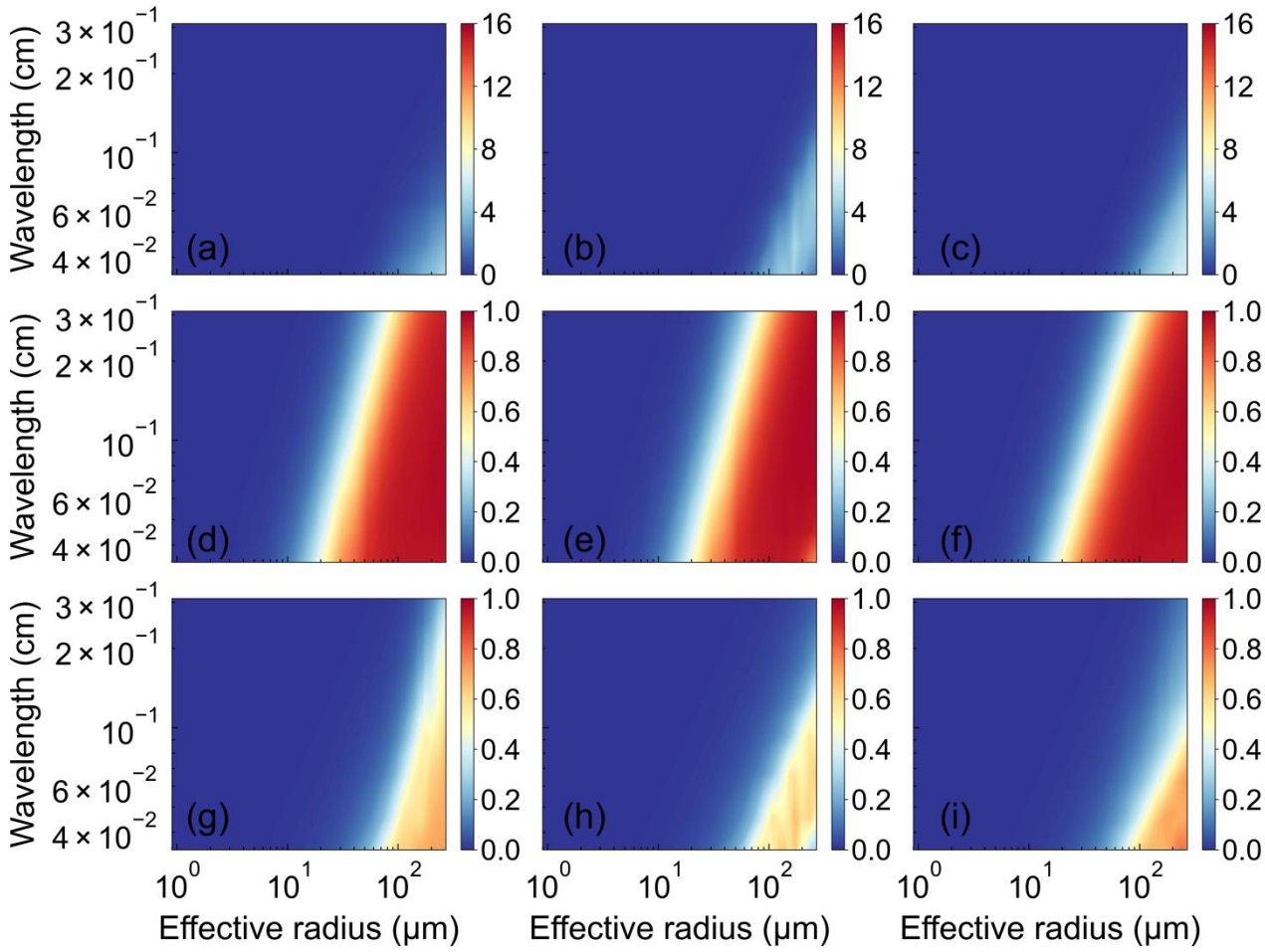

**Figure 3:** The comparison of the extinction efficiency, single-scattering albedo and asymmetry factor as functions of 31 particle sizes (1-270 μm) and 20 terahertz wavelengths (0.03-0.3 cm) for the (a, d, g) Voronoi, (b, e, h) Sphere and (c, f, i) Column ICS models.

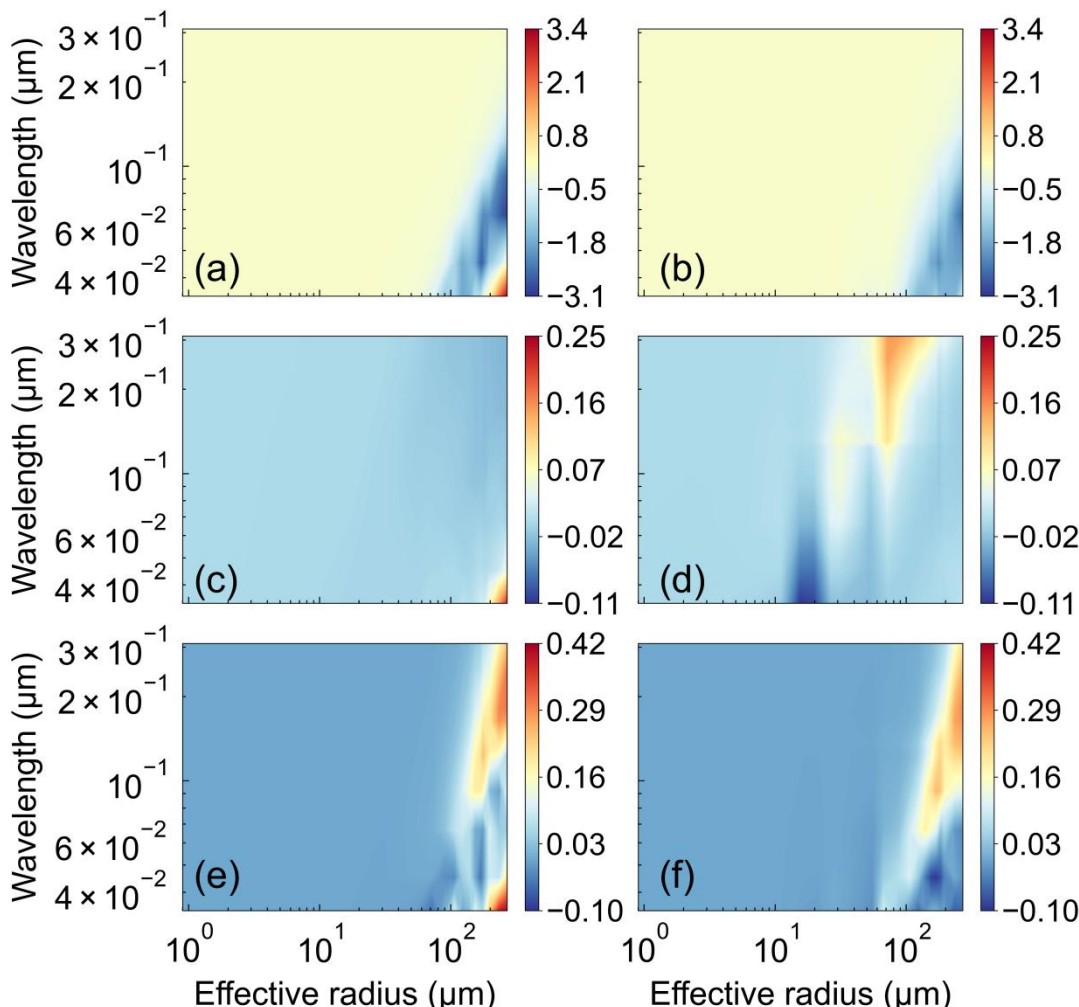

**Figure 4: The (a, c, e) Voronoi minus Sphere and the (b, d, f) Voronoi minus Column ICS model differences in (top row) extinction efficiency, (middle row) single-scattering albedo and (bottom row) asymmetry factor as functions of 31 particle sizes (1-270 μm) and 20 terahertz wavelengths (0.03-0.3 cm).**


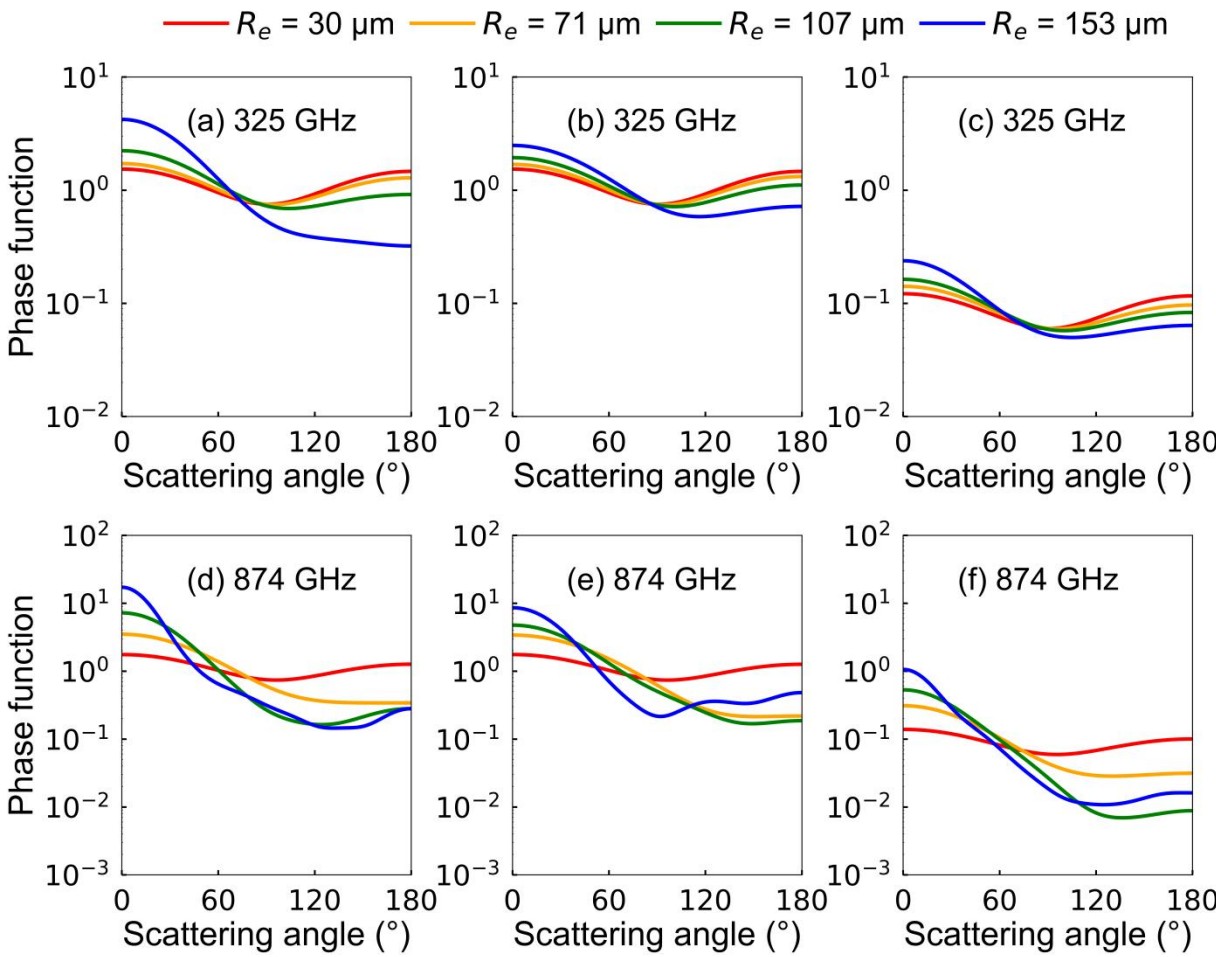

**Figure 5: The scattering phase functions for ice particles with four sizes ($R_e$ = 30, 71, 107 and 153 μm) for the (a, d) Voronoi, (b, e) Sphere and (c, f) Column ICS models with a refractive index of 1.78 + 0.005$i$ in the 325 GHz and 1.78 + 0.015$i$ in the 874 GHz, respectively.**

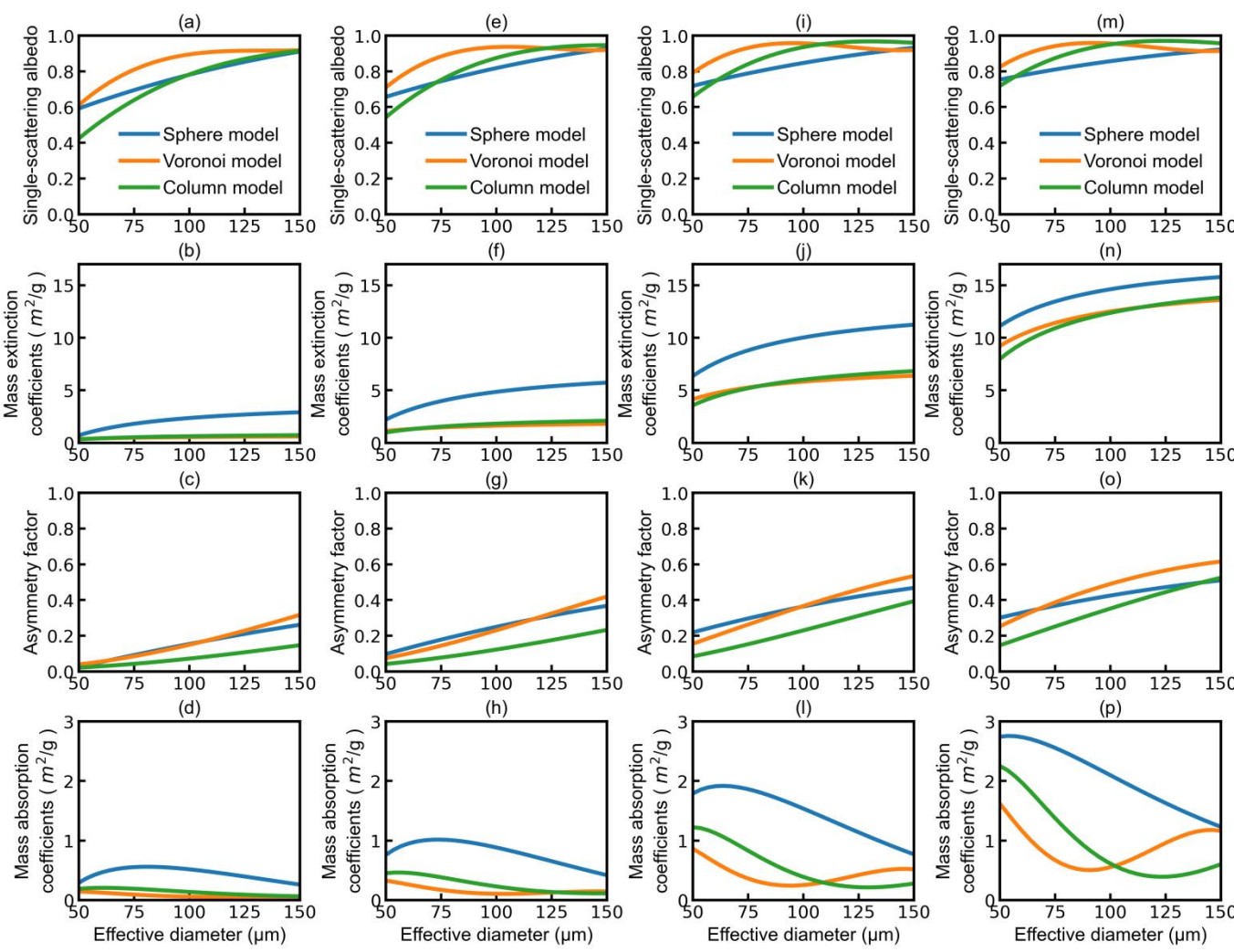


**Figure 6: The comparison of the parameterized single-scattering albedo, mass extinction coefficients, asymmetry factor and mass absorption coefficients as functions of effective diameters for 325, 448, 664 and 874 GHz for the Sphere (blue line), Voronoi (orange line) and Column (green line) ICS models.**

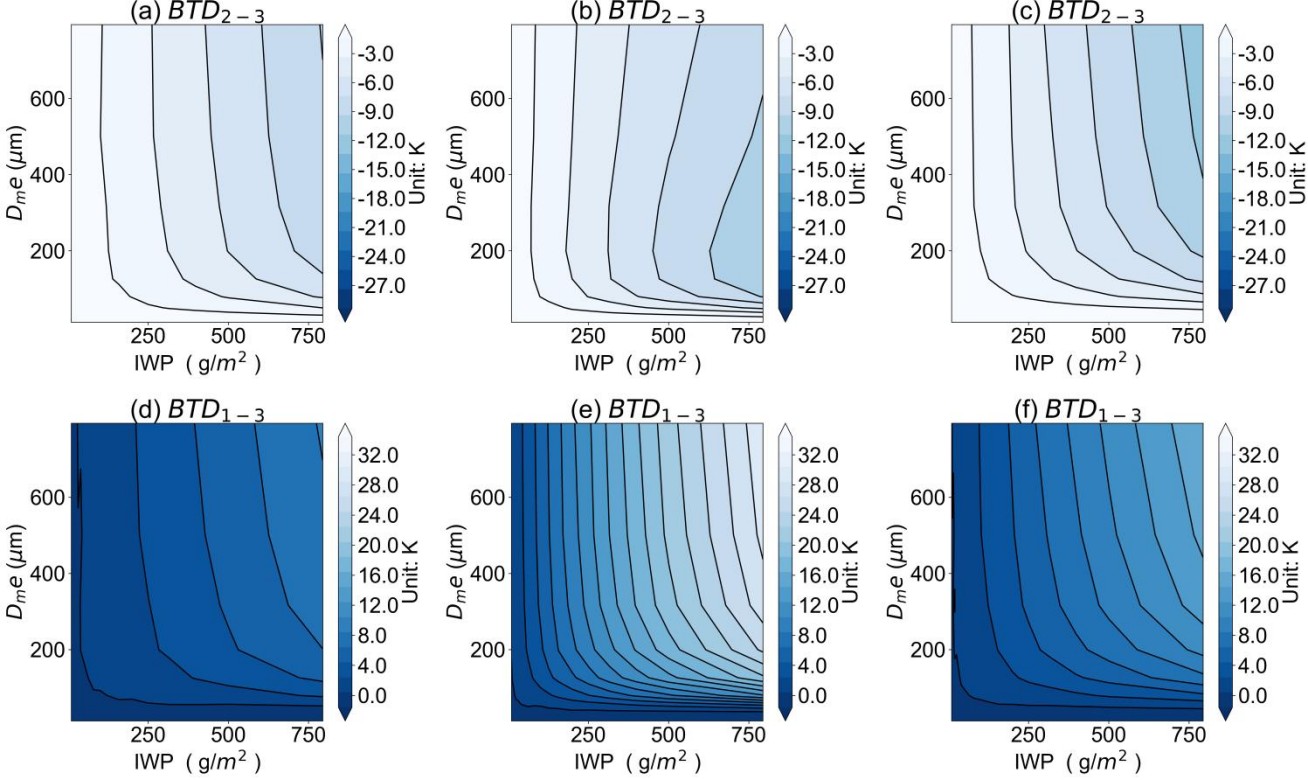

**Figure 7: The BTD$_{2-3}$ and BTD$_{1-3}$ for the (a, d) Voronoi, (b, e) Sphere and (c, f) Column ICS models as functions of the IWP and $D_{me}$, respectively.**

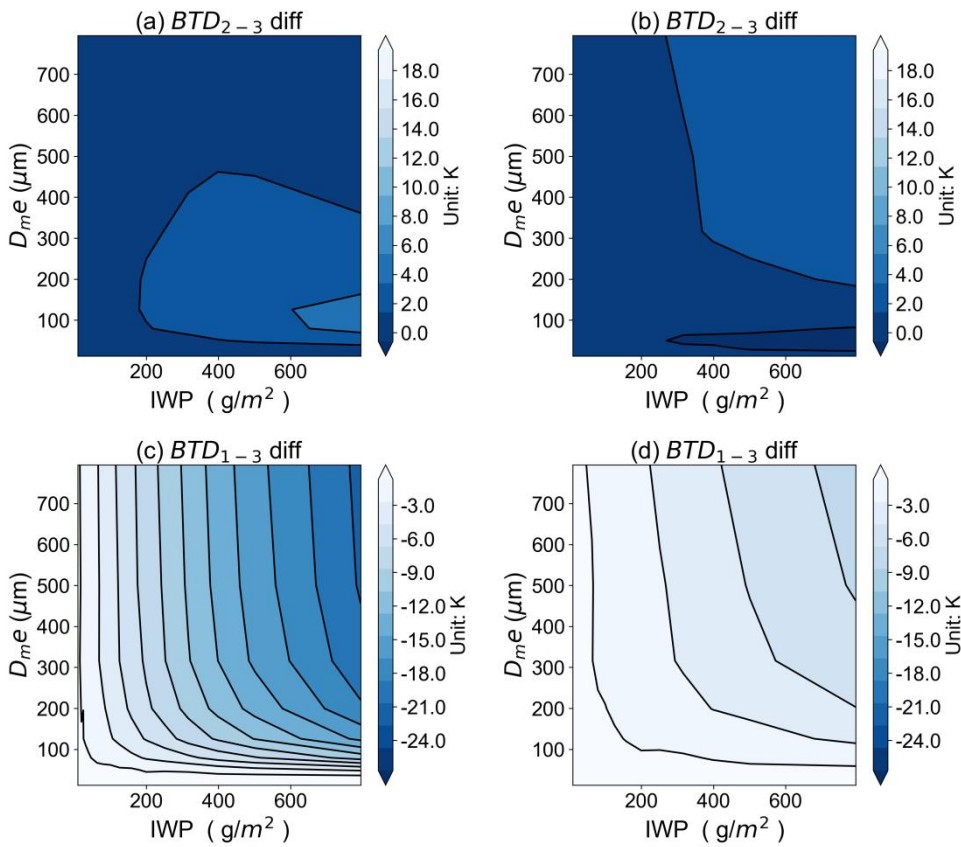


**Figure 8: The BTD$_{2\text{-}3}$ and BTD$_{1\text{-}3}$ differences for the (a, c) Voronoi minus Sphere and (b, d) Voronoi minus Column ICS models as functions of the IWP and $D_{me}$, respectively.**

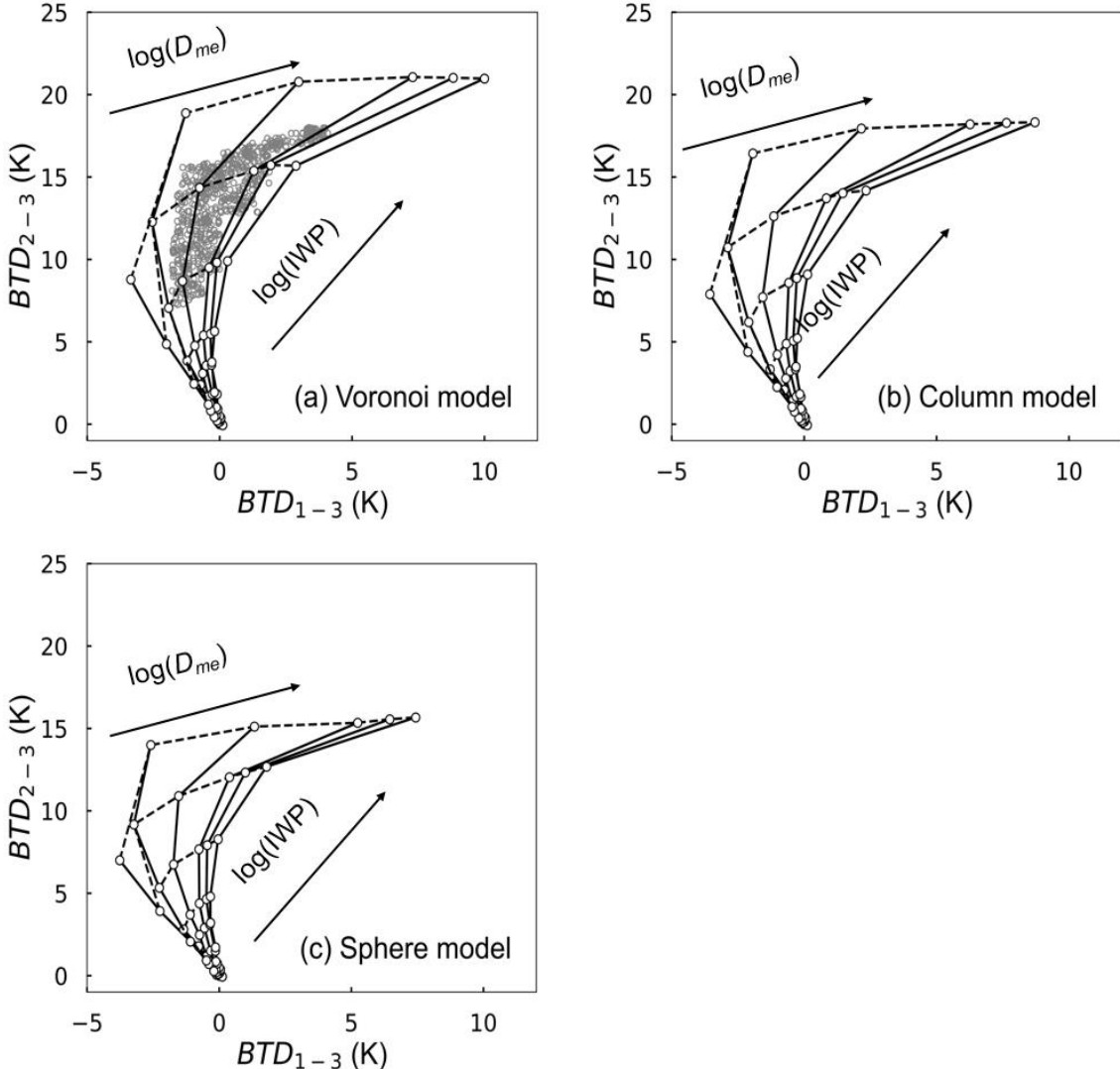

**Figure 9: The LUT of BTD$_{2-3}$ and BTD$_{1-3}$ for the (a) Voronoi, (b) Column and (c) Sphere ICS models varying with the logarithm of IWP and $D_{me}$. Grey dots in circles represent the randomly generated 2000 test data from the RSTAR model.**

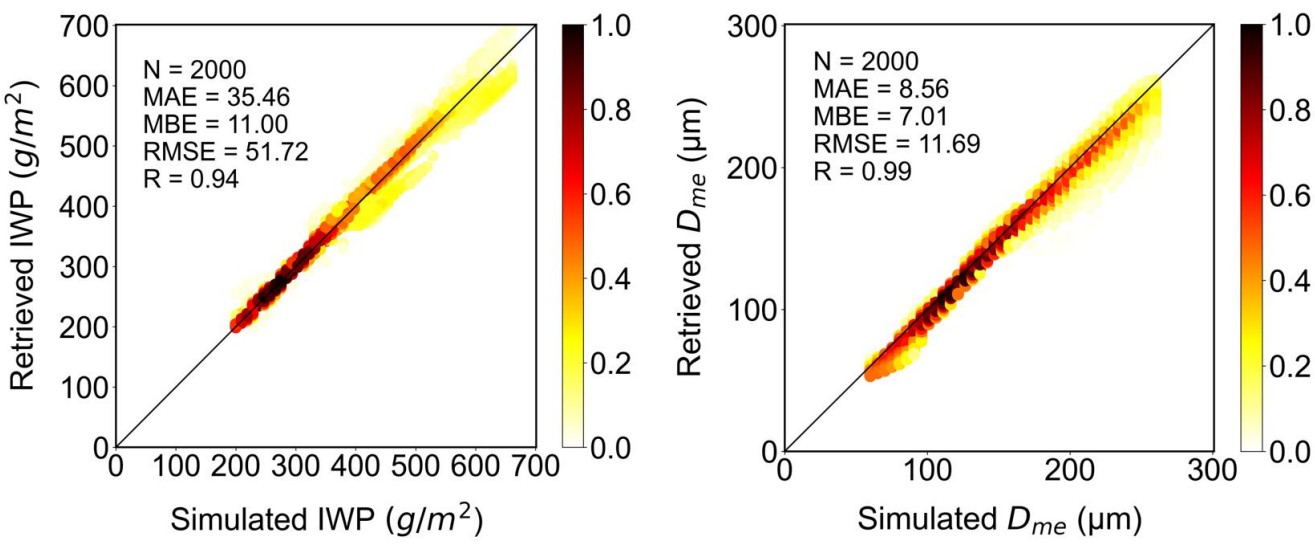

**Figure 10: The scatterplots of the randomly generated 2000 test data and the retrieved results. The left panel and the right panel are for the IWP and $D_{me}$, respectively.**


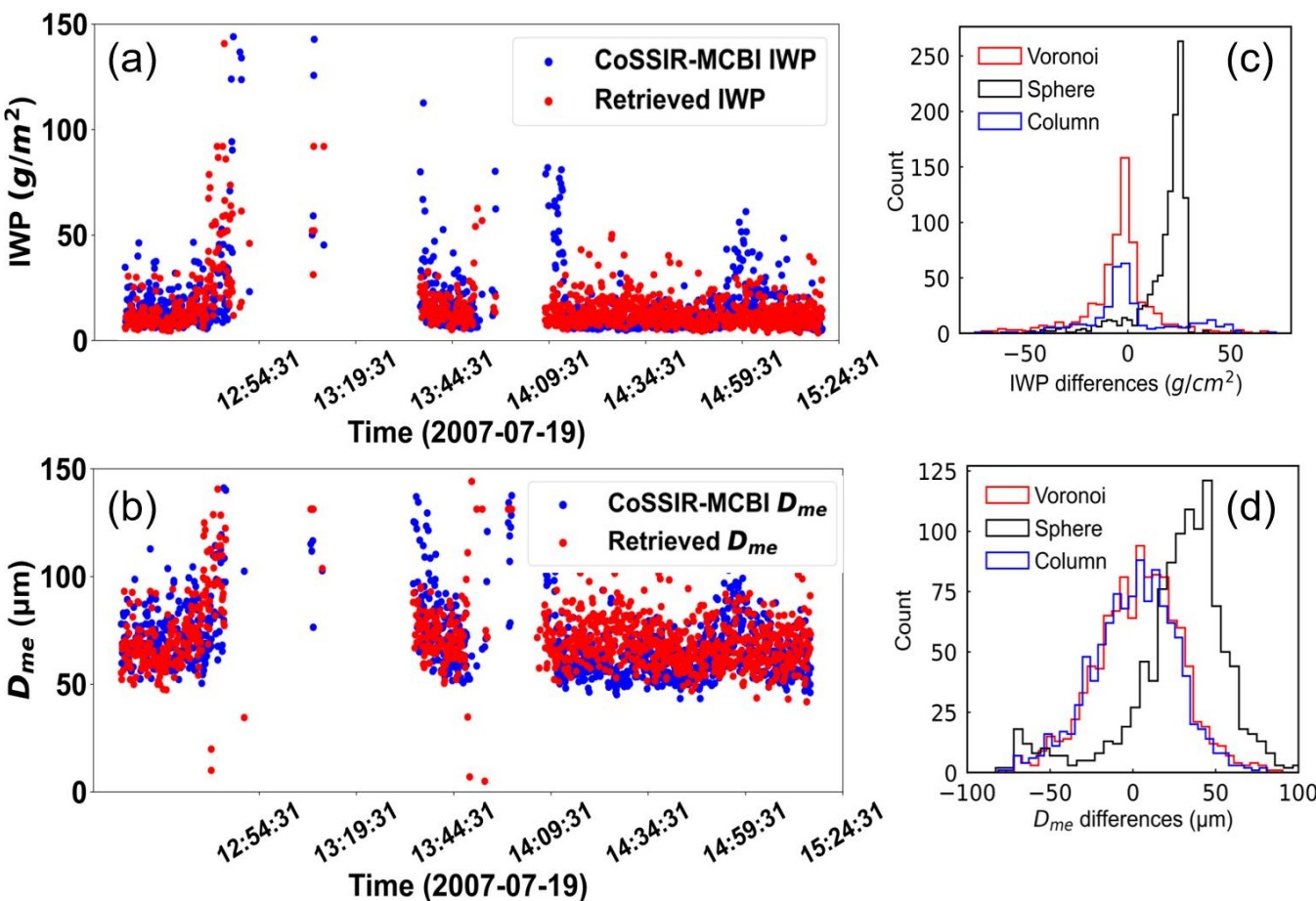

Figure 11: The scattered red dots are the retrieved (a) IWP and (b) $D_{me}$ (bottom row) validated by the results from the CoSSIR-MCBI algorithm (blue scattered dots). The joint histogram of differences of the retrieved (c) IWP and (d) $D_{me}$ between the Voronoi, Sphere, Column schemes and the CoSSIR-MCBI results, respectively.

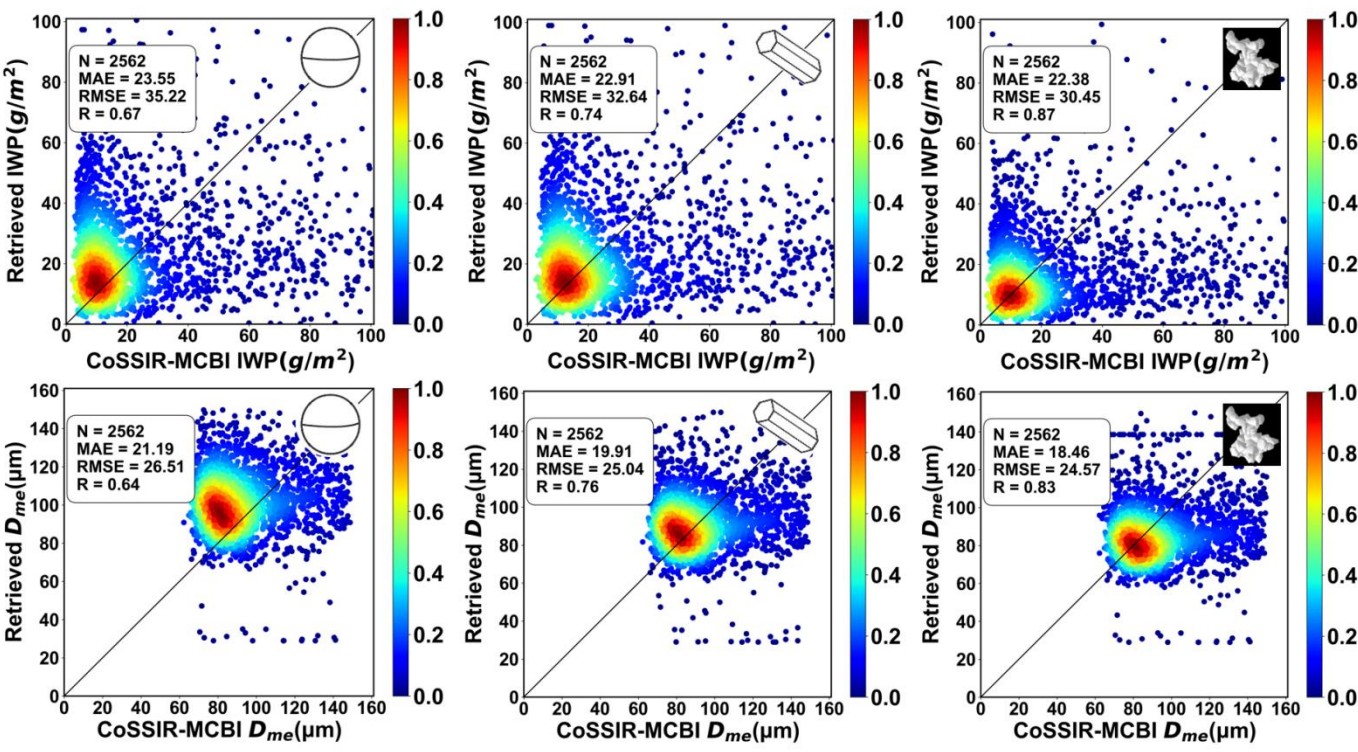

**Figure 12: The scatterplots of the retrieved IWP (top row) and $D_{me}$ (bottom row) against the CoSSIR-MCBI results**
**for the Sphere (right column), Column (middle column) and Voronoi (left column) ICS models.**