# Peer review of "Retrieval of Terahertz Ice Cloud Properties from airborne measurements based on the irregularly shaped Voronoi ice scattering models"

_Atmospheric Measurement Techniques, 2022_

## Referee Comment (RC1)

Full title: Retrieval of Terahertz Ice Cloud Properties from airborne measurements based on the irregularly shaped Voronoi ice scattering models Authors: Li et al.

This paper investigated the ability of Sphere and Voronoi model in retrieving cloud microphysical properties such as ice water path (IWP) and effective particle radius (Re) using airborne measurements. Sensitivity results indicate that TOC BTDs between 640 and 874 GHz is used for IWP, while BTDs between 380, 640 and 874 GHz is used for Re. In addition, retrieved results of IWP and Re from Voronoi model are better than that of the Sphere model compared with airborne ones. Overall, this manuscript is clear. However, there are several issues that need to be taken care of before this paper becomes acceptable for publication.

Specific comments:

- How about the previous research in terahertz band? In the introduction, I only saw Li's research (Li et al., 2016). How about the accuracy of retrieved IWP and Re of previous studies using different ICS models, like aggregates, hollow columns, flat plates, rosettes and spheres?
- 2. The "Inversion results" part is too short, and the results and validation sections are not insightful. You simply present the validation metrics like MBE, RMSE, and R, etc. Why is the Voronoi better than the sphere model?
- For Figure 7, 2000 test data were generated by the RTSRA and plotted on the Figure
  7 with black dots, why are there only 19 points?
- 4. Why do the problems of 1 and 2 in the figure occur, see below?

5. Increase the drawing range of Y in Figure 10, from currently  $0 \sim 145$ , to  $0 \sim 160$ . I

want to see the sphere have the same horizontal line problem.

---

## Referee Comment (RC2)

Review of "Retrieval of Terahertz Ice Cloud Properties from airborne measurements based on the irregularly shaped Voronoi ice scattering models" by Ming Li et al.

General comments

The paper is about applying the Voronoi model to the retrieval of IWP and $r_e$ using brightness temperature differences between 380, 640, and 874 GHz. Not surprisingly, the authors find the Voronoi model re-produces previous retrievals of IWP and $r_e$ more accurately than the sphere. This aspect is not new in the microwave and sub-millimetre, see for instance, the study by Eriksson et al. (2015), https://amt.copernicus.org/articles/8/1913/2015/amt-8-1913-2015.pdf as to why the authors find the sphere to be an inadequate representation of non-spherical ice scattering in the microwave and sub-mm regions. The important aspect of this paper is that the Voronoi model has been previously applied to simulate solar and infrared observations, and now it is being applied over the Terahertz region to see how well the model performs there. However, as to how skillfully it performs against other ice crystal models is yet to be tested.  The authors find very good correlations between the Voronoi-based retrievals and Evan's Bayesian retrievals using data from the CoSSIR instrument. The paper is relatively well-written and can be followed. The figures are also well represented, and the analysis is quantitative, with no obvious flaws. Further proof-reading is recommended to help improve the flow of the paper. This paper could be significantly improved, which if followed, would make the paper a more important contribution to the remote sensing of ice cloud in the microwave and sub-millimetre regions of the spectrum.

The major recommendations are as follows:

1. It is felt that the authors missed an opportunity to test the veracity of the Voronoi model in the microwave and sub-mm by not comparing their results with another more representative ice crystal scattering model. For instance, why not use the scattering models contained in the ARTS database of single-scattering properties? One model from the ARTS collection of models to try and test against the Voronoi model is the large column aggregate model. This model was shown by Fox (2020) to simulate better than some of the other models, the microwave and sub-millimeter brightness temperature measurements between the frequencies of 183 and 664 GHz. I recommend the authors compare their retrievals and simulations against more realistic ice crystal scattering models such as the ARTS large column aggregate. See, Fox, S. An Evaluation of Radiative Transfer Simulations of Cloudy Scenes from a Numerical Weather Prediction Model at Sub-Millimetre Frequencies Using Airborne Observations. Remote Sens. 2020, 12, 2758. https://doi.org/10.3390/rs12172758.

2. The authors make use of existing retrievals of $r_e$ and IWP to test the Voronoi model but do not make use of the independent measures of IWP and $r_e$ as derived from the in-situ aircraft during TC4. Why is this? Is the in-situ aircraft data not available?  Was there no in-situ data co-incident with the radiometric measurements? The problem with comparing with the existing CoSSIR retrievals is that those retrievals are based on differing assumptions of mass, ice crystal shape and PSDs – comparing apples and oranges. It could be said that the CoSSIR ice crystal shape and mass assumptions are just as valid as the Voronoi model, yet they may be entirely different. It would be much better to compare retrievals with in-situ measures if those are available.

3. The authors propose a convoluted and unnecessary method of relating $r_e$ to $D_{me}$. This is surprising, since in the terahertz region the scattering cross sections are more dependent on mass rather than area. Why use an area-weighted size such as re rather than a mass-weighted size such as $D_{me}$? The problem with using $r_e$ in the terahertz region is nicely explained in the study by Seiron et al. (2017), see

. In the region of interest, a mass-weighted size would be the more appropriate characteristic size of the PSD to utilize in this paper.

4. No evidence is presented as to how representative the PSDs used in the analysis are for the TC4 cases considered in the paper. The best way to do this is to derive the moments of the assumed PSDs and in-situ PSDs and show how well correlated they are. Of course, if the in-situ PSDs are not available, this cannot be done!

5. Related to 4, is the question of how representative is the ERA5 re-analysis product for a couple of TC4 cases? The temperature, water vapour and ozone profiles are important in the radiative transfer simulations. If the ERA5 re-analysis product is not representative of the actual state of the atmosphere for those few days, this could bias the brightness temperature difference results. The authors should compare some of the ERA5 atmospheric profiles with the aircraft profiles, if the latter are available.

6. The authors need to provide images of the Voronoi model with increasing ice crystal size, such that it can be seen by readers how the model aggregation varies with size. What are also required in the revised paper are the Voronoi model's mass– and area–dimension power laws. These power law relations will go some way to explaining the single-scattering results and sensitivities of the Voronoi model to IWP and the characteristic size of the PSD in the brightness temperature difference sensitivity analysis. The fractal dimensions of mass and area of the Voronoi model are important in these respects.

7. Apart from plotting the retrieved quantities, a further measure of how well the Voronoi model represents the measured brightness temperatures at the three channels is to plot the residuals (i.e. brightness temperature differences between the forward model and measurements) as a function of time for all three channels.

The minor comments are listed as follows with page numbers:

1. Introduction line 34. Since the authors discuss 20-30% of the global cloud mass, would it not be better to cite more updated studies that more directly measure the ice mass such as studies using CloudSat global retrievals of ice mass? As well as mm-wave retrievals of ice mass?

2. Line 36. As the paper is discussing Terahertz frequencies, another important property of large ice crystals that contribute to the radiative properties of ice cloud is their orientation.

3. List of citations on line 47. Fox (2020) should be added to this list?

4. Line 51. The description of Fox et al. (2019) needs to be more accurate, the study also used sub-mm frequencies up to 664 GHz, and in Fox (2020). The works of Fox (2019,2020) includes the Terahertz region, and not just the microwave.

5. Line 79, again ice crystal orientation is also an important consideration here.

6. Line 97. Another numerical method that could be included in this list is the Boundary Element Method, which has recently been applied to very complex ice crystals by Kleanthous et al. (2022): Antigoni Kleanthous, Timo Betcke, David P. Hewett, Paul Escapil-Inchauspé, Carlos Jerez-Hanckes, Anthony J. Baran, Accelerated Calderón preconditioning for Maxwell transmission problems, Journal of Computational Physics, Volume 458, 2022, 111099, ISSN 0021-9991, https://doi.org/10.1016/j.jcp.2022.111099. A further paper here could be Mano

(2000), who applied BEM to hexagonal ice columns. "Exact solution of electromagnetic scattering by a three-dimensional hexagonal ice column obtained with the boundary-element method," Appl. Opt. **39**, 5541-5546.

7. Line 98. This GOA acronym has not been defined - should it be GOM?

8. The discussion beginning on line 108. Another ICS model worthy of note in this context is the ensemble model of cirrus ice crystals developed by Baran and Labonnote (2007). The ensemble model attempts to be more representative of the evolution of the ice crystal aggregation process as a function of increasing size, see Baran, A.J. and Labonnote, L.-C. (2007), A self-consistent scattering model for cirrus. I: The solar region. Q.J.R. Meteorol. Soc., 133: 1899-1912. ) https://doi.org/10.1002/qj.164), and Baran et al. 2014 (Baran, A.J., Cotton, R., Furtado, K., Havemann, S., Labonnote, L.-C., Marenco, F., Smith, A. and Thelen, J.-C. (2014), A self-consistent scattering model for cirrus. II: The high and low frequencies. Q.J.R. Meteorol. Soc., 140: 1039-1057. https://doi.org/10.1002/qj.2193).

9. Typo on line 117 Mo.,del -> Model

10. Line 118. The word effectiveness is sufficient, using the word "superiority" is inappropriate here because it has not been proven relative to all other models that are now available.

11. Line 122. ICI is not correct here, the instrument is ISMAR (International Sub-Millimeter wave Airborne Radiometer) described in Fox et al., 2017. ISMAR was jointly funded by the Met Office and ESA – not ICI.

12. Section 2.1. Which refractive indices are being used to compute the SSPs? The refractive indices in the microwave and sub-millimeter are temperature dependent - is this dependence considered in the simulations that follow? If not, which temperature has been assumed in the calculation of the SSPs? How have you justified the selection of this temperature?

13. Section 2.3. Is the cloud between the boundaries assumed to be homogeneous? If so, please state this.

14. Equation 2, line 186. In the denominator, this is why you need to provide the model's mass–dimension relationship.

15. Equation 4, line 205. In the denominator, this is why you need to provide the model's area–dimension relationship.

16. Equation 7, there is a missing wavelength dependence in the denominator for the scattering cross section.

17. Equations 9 -12, how accurate are the parametric fits as a function of $D_e$?

18. Figure 9, this figure might be better plotted as a PDF of the retrievals, and statistically measure how different the distributions are from the reference PDFs using some statistical measure.

---

## Referee Comment (RC5)

Review for "Retrieval of Terahertz Ice Cloud Properties from airborne measurements based on the irregularly shaped Voronoi ice scattering models"

Authors: Li et al.

General comments

This study compares the capability of the Voronoi and sphere models in the retrieval of IWP and $r_e$ using aircraft-based terahertz measurements. The study shows that the Voronoi model can provide promising results as compared to Evan's Bayesian retrievals using data from the CoSSI instrument. The inversion algorithm among the Voronoi and Sphere models suggests that the Voronoi model is better than the Sphere model. The paper seems clear and well-written. From the single-scattering properties of ice particles to the ice cloud retrievals, the structure is complete, and the analysis is quantitative. In my opinion, this paper could be a good supplement to the development of ice cloud terahertz remote sensing. The topic presented in this study is suitable for Atmospheric Measurement Techniques. I recommend Minor Revisions for publication.

Specific comments

1. Are those comparisons between the single-scattering properties of the Voronoi and Sphere models under the same complex refractive index of ice particles? Please add the real and imaginary parts of the refractive index at 325 and 874 GHz in Figures 2 and 4.

2. In the paper, the $BTD_{1-3}$ may be confused with the $BTD_{1-2}$-$BTD_{2-3}$. Please confirm the Acronyms throughout the manuscript.

3. In Figure 6, the brightness temperature differences at 640GHz are shown, albeit not used in the following retrieval. Please give explanations or redraw Figure 6.

4. I recommend the authors give more possible explanations about why large difference exists for large ice particles in the result section.

---

## Author Comment (AC1)

Comment on amt-2022-247

Anonymous Referee # 2

Referee comment RC1 on "Retrieval of Terahertz Ice Cloud Properties from airborne measurements based on the irregularly shaped Voronoi ice scattering models" by Ming Li et al.

**General comments**

This paper investigated the ability of Sphere and Voronoi model in retrieving cloud microphysical properties such as ice water path (IWP) and effective particle radius (Re) using airborne measurements. Sensitivity results indicate that TOC BTDs between 640 and 874 GHz is used for IWP, while BTDs between 380, 640 and 874 GHz is used for Re. In addition, retrieved results of IWP and Re from Voronoi model are better than that of the Sphere model compared with airborne ones. Overall, this manuscript is clear. However, there are several issues that need to be taken care of before this paper becomes acceptable for publication.

Response: Thank you very much for your significant comments.

**Specific comments**

1. How about the previous research in terahertz band? In the introduction, I only saw Li's research (Li et al., 2016). How about the accuracy of retrieved IWP and Re of previous studies using different ICS models, like aggregates, hollow columns, flat plates, rosettes and spheres?

Response: Thanks for the comments. We have added illustrations about the result accuracy of previous studies in the terahertz band in section 1 as shown below.

Lines 76-81: "For the accuracy of the BMCI method, validation results stated that for clouds with IWP greater than 5 g/m$^2$ the overall median retrieval error is about 30% for IWP and 15% for $D_{me}$ (Evans et al., 2002). Jimenez et al. (2007) used the neural network method to retrieve the IWP and $D_{me}$. Results showed overall median relative errors of around 20% for IWP and 33 μm for $D_{me}$ for a mid-latitude winter scenario, and 17% for IWP and 30 μm for $D_{me}$ for a tropical scenario. Based on these studies,

Buehler et al. (2007) proposed a formal scientific mission requirement for a passive submillimeter-wave cloud ice mission based on the background and early research. The requirements are the low IWP should be less than 10 g/m$^2$, the high ice water path should be less than 50%, and the particle diameter should be less than 50 μm. Lately, Liu et al. (2021) proposed an inversion method for the remote sensing of ice clouds at terahertz wavelengths based on a genetic algorithm. Results showed the absolute error of the low IWP (below 20 g/m$^2$) is small, while the relative error of the high IWP is generally maintained at around 10%, and the absolute error of the effective particle diameter is mostly around 4 μm."

2. The "Inversion results" part is too short, and the results and validation sections are not insightful. You simply present the validation metrics like MBE, RMSE, and R, etc. Why is the Voronoi better than the sphere model?

Response: Thanks for the comments. In this study, we have added more analysis and explanations of the results in terms of the differences in the single scattering properties of three ice crystal models. The following description is added at the end of section 4.4.

Lines 323-328: "According to the sensitivity results of Figures 6 and 7, the Voronoi ICS scheme has higher BTD$_{2\text{-}3}$ and BTD$_{1\text{-}3}$ compared to the Sphere and Column ICS schemes, especially for large particles and IWP. This characteristic is also shown in Figure 8. This can be explicitly explained by the larger asymmetry factor of the Voronoi ICS model compared to the Sphere and Column ICS models. Thus, stronger forward scattering energy can be detected for the Voronoi ICS model than the other two models. The look-up table of the Voronoi ICS model can cover more IWP and $D_{me}$. The brightness temperature variations of the Voronoi-shaped ice clouds are more prominent and sensitive to the IWP and $D_{me}$. Therefore, the results of the Voronoi ICS model are better than the other two models."

3. For Figure 7, 2000 test data were generated by the RTSRA and plotted on the Figure 7 with black dots, why are there only 19 points?

Response: The 19 black dots shown in Figure 7 are only used to generally indicate that the selected test dataset is within the coverage of the look-up table. According to

the comments, we have presented all 2000 test points using grey dots in Figure 7 as shown below.

[Figure]

Figure 7: The LUTs of BTD$_{2-3}$ and BTD$_{1-3}$ for the (a) Voronoi, (b) Column and (c) Sphere ICS models varying with the logarithm of IWP and $D_{me}$. Grey dots in circles represent the randomly generated 2000 test data from RSTAR model.

4. Why do the problems of 1 and 2 in the figure occur, see below?

[Figure]

645   Figure 10: The scatterplots of the retrieved IWP (top row) and $D_{me}$ (bottom row) against the CoSSIR-MCBI results for the Sphere (right column) and Voronoi (left column) ICS models.

Response: Thanks. According to the look-up table as shown above, there are overlapping lines when $D_{me}$ is small ($D_{me}$ < 40 μm) and large ($D_{me}$ > 140 μm). When

BTD$_{2\text{-}3}$ and BTD$_{1\text{-}3}$ data fall under such overlapping lines of the look-up table, this overlapping region can lead to obtaining the same IWP and $D_{me}$ when searching the look-up table. That is why the "horizontal line" problem occurs for $D_{me} < 40$ μm and $D_{me} > 140$ μm.

5. Increase the drawing range of Y in Figure 10, from currently 0~145, to 0~160. I want to see the sphere have the same horizontal line problem.

Response: According to the comments, we have increased the range of $D_{me}$ from 0~145, to 0~160, and redrawn Figure 10 as follows. As shown below, the "horizontal line" problem does not exist for the Sphere and Column ICS models when $D_{me} > 40$ μm. That is because BTD$_{2\text{-}3}$ and BTD$_{1\text{-}3}$ do not fall where the lines overlap in the look-up tables for the Sphere and Column ICS models when $D_{me} > 40$ μm.

[Figure]

Figure 10: The scatterplots of the retrieved IWP (top row) and $D_{me}$ (bottom row) against the CoSSIR-MCBI results for the Sphere (right column), Column (middle column) and Voronoi (left column) ICS models.

Reference:
Buehler, S., Jimenez, C., Evans, K., Eriksson, P., Rydberg, B., Heymsfield, A., Stubenrauch, C., Lohmann, U., Emde, C., John, V., Tr, S., and Davis, C.: A concept for a satellite mission to measure cloud ice water path and ice particle size, Q J Roy Meteor Soc, 133, 109-128, 10.1002/qj.143, 2007.
Evans, K. F., Walter, S. J., Heymsfield, A. J., and McFarquhar, G. M.: Submillimeter-Wave Cloud Ice Radiometer: Simulations of retrieval algorithm

performance, Journal of Geophysical Research: Atmospheres, 107, AAC 2-1-AAC 2-21, doi:10.1029/2001JD000709, 2002.

Jimenez, C., Buehler, S., Rydberg, B., Eriksson, P., and Evans, K.: Performance simulations for a submillimetre wave cloud ice satellite instrument, Q J Roy Meteor Soc, 133, 129-149, 10.1002/qj.134, 2007.

Liu, L., Weng, C., Li, S., Letu, H., Hu, S., and Dong, P.: Passive Remote Sensing of Ice Cloud Properties at Terahertz Wavelengths Based on Genetic Algorithm, Remote Sensing, 13, 735, doi:10.3390/rs13040735, 2021.

---

## Author Comment (AC2)

Comment on amt-2022-247

Anonymous Referee # 3

Referee comment RC2 on "Retrieval of Terahertz Ice Cloud Properties from airborne measurements based on the irregularly shaped Voronoi ice scattering models" by Ming Li et al.

**General comments**

The paper is about applying the Voronoi model to the retrieval of IWP and re using brightness temperature differences between 380, 640, and 874 GHz. Not surprisingly, the authors find the Voronoi model re-produces previous retrievals of IWP and re more accurately than the sphere. This aspect is not new in the microwave and sub-millimetre, see for instance, the study by Eriksson et al. (2015), https://amt.copernicus.org/articles/8/1913/2015/amt-8-1913-2015.pdf as to why the authors find the sphere to be an inadequate representation of non-spherical ice scattering in the microwave and sub-mm regions. The important aspect of this paper is that the Voronoi model has been previously applied to simulate solar and infrared observations, and now it is being applied over the Terahertz region to see how well the model performs there. However, as to how skillfully it performs against other ice crystal models is yet to be tested. The authors find very good correlations between the Voronoi-based retrievals and Evan' s Bayesian retrievals using data from the CoSSIR instrument. The paper is relatively well-written and can be followed. The figures are also well represented, and the analysis is quantitative, with no obvious flaws. Further proof-reading is recommended to help improve the flow of the paper. This paper could be significantly improved, which if followed, would make the paper a more important contribution to the remote sensing of ice cloud in the microwave and sub-millimetre regions of the spectrum.

Response: Thank you very much for your significant comments.

**Major comments**

1. It is felt that the authors missed an opportunity to test the veracity of the Voronoi model in the microwave and sub-mm by not comparing their results with another more representative ice crystal scattering model. For instance, why not use the scattering models contained in the ARTS database of single-scattering properties? One model from the ARTS collection of models to try and test against the Voronoi model is the large column aggregate model. This model was shown by Fox (2020) to simulate better than some of the other models, the microwave and sub-millimeter brightness temperature measurements between the frequencies of 183 and 664 GHz. I recommend the authors compare their retrievals and simulations against more realistic ice crystal scattering models such as the ARTS large column aggregate. See, Fox, S. An Evaluation of Radiative Transfer Simulations of Cloudy Scenes from a Numerical Weather Prediction Model at Sub-Millimetre Frequencies Using Airborne Observations. Remote Sens. 2020, 12, 2758. https://doi.org/10.3390/rs12172758.

Response: Thanks for the comments. We have added the hexagonal column ice crystal scattering (ICS) model from the ARTS collection of models to compare with our results in this study. We also cited the document of Fox (2020) in section 1 as follows. Lines 85-87: "Fox (2020) found that the randomly-oriented large column aggregate can simulate observed brightness temperatures between 183 and 664 GHz with high accuracy."

The descriptions of the hexagonal column ICS model are added in section 2.1 as shown below.

Lines 149-151: "The randomly-oriented hexagonal column (referred to as the Column hereafter) ICS model was defined by Yang et al. (2000a). Their aspect ratios $a/L$ (defined as the ratio of the semiwidth $a$ of a particle to its length $L$) of Column ICS models are defined as 0.35 and 3.48 respectively when $L$ is less than 100 μm and greater than or equal to 100 μm. The single-scattering property database of the Column ICS model used in the study is developed by Hong (2007, 2009) using the discrete dipole approximation method at frequencies of 100-1000 GHz."

2. The authors make use of existing retrievals of re and IWP to test the Voronoi model but do not make use of the independent measures of IWP and re as derived from the in-situ aircraft during TC4. Why is this? Is the in-situ aircraft data not available? Was there no in-situ data co-incident with the radiometric measurements? The problem with comparing with the existing CoSSIR retrievals is that those retrievals are based on differing assumptions of mass, ice crystal shape and PSDs – comparing apples and oranges. It could be said that the CoSSIR ice crystal shape and mass assumptions are just as valid as the Voronoi model, yet they may be entirely different. It would be much better to compare retrievals with in-situ measures if those are available.

Response: Thanks for the comments. As you mentioned, the main reason is that there was no in-situ measurement data of IWP and re co-incident with the radiometric measurements.

3. The authors propose a convoluted and unnecessary method of relating re to Dme. This is surprising, since in the terahertz region the scattering cross sections are more dependent on mass rather than area. Why use an area-weighted size such as re rather than a mass-weighted size such as Dme? The problem with using re in the terahertz region is nicely explained in the study by Seiron et al. (2017), see https://agupubs.onlinelibrary.wiley.com/doi/full/10.1002/2017JD026494. In the region of interest, a mass-weighted size would be the more appropriate characteristic size of the PSD to utilize in this paper.

Response: Thanks. In this study, the ice crystal size parameter entered into the radiative transfer model (RSTAR) used in our study is the effective model radius $R_e$ rather than mass-weighted particle size $D_{me}$. And the $R_e$ and $D_{me}$ are defined according to the following formula,

$$R_e = \frac{\int_0^\infty r^3 n(r) dr}{\int_0^\infty r^2 n(r) dr} \, , \tag{1}$$

$$D_{me} = \frac{\int_0^\infty D m(D) n(D) dD}{\int_0^\infty m(D) n(D) dD} \, , \tag{2}$$

where $r$ is the equivalent-volume sphere's particle radius, $D$ is the maximum particle dimension of ice particles, $m$ is the particle mass, $n$ is the particle size distribution. Hence, we developed a conversion relationship between $R_e$ (independent

variable) and $D_{me}$ (dependent variable) combined with different particle size distributions. Based on this relationship, we have unified all the $R_e$ into $D_{me}$ in the revised manuscript.

4. No evidence is presented as to how representative the PSDs used in the analysis are for the TC4 cases considered in the paper. The best way to do this is to derive the moments of the assumed PSDs and in-situ PSDs and show how well correlated they are. Of course, if the insitu PSDs are not available, this cannot be done!

Response: As you mentioned, the main reason is that the in-situ PSDs during the TC4 mission are not available. However, we use in-situ measurements of PSD data (Heymsfield et al., 2013) from 11 field programs spanning a wide range of locations (ranging from 12°S to 70°N latitudes and from 148°W to 130°E longitudes) and encompassing the temperature range 0° to -86°C, and with altitudes from near the surface to 18.7 km. This dataset is representative of the wide range of conditions where ice clouds are found in the troposphere and lower stratosphere on a near-global scale (Li et al., 2022; Heymsfield et al., 2013). Relationships expressing PSDs and the maximum particle dimensions are presented in terms of their temperature, as shown below (Figure 1). The relationships developed can serve as a basis for developing reliable parameterizations.

[Figure]

Figure 1. Ice cloud particle size distributions for different temperatures.

5. Related to 4, is the question of how representative is the ERA5 re-analysis product for a couple of TC4 cases? The temperature, water vapour and ozone profiles are important in the radiative transfer simulations. If the ERA5 re-analysis product is not representative of the actual state of the atmosphere for those few days, this could bias the brightness temperature difference results. The authors should compare some of the ERA5 atmospheric profiles with the aircraft profiles, if the latter are available.

Response: As you mentioned, the main reason is that the in-situ aircraft atmospheric profiles during the TC4 mission are unavailable. However, many documents have verified the high accuracy of the ERA5 products. For example, Graham et al. (2019) evaluated five atmospheric reanalyses, including ERA5, ERA-Interim, Japanese 55-year Re-Analysis (JRA-55), Climate Forecasting System Reanalysis-version 2 (CFSv2), and Modern Era Retrospective analysis for Research and Applications-version 2 (MERRA-2) using observations from 50 radiosondes. Overall, the newly released ERA5 has higher correlation coefficients than any other reanalysis (Graham et al., 2019).

6. The authors need to provide images of the Voronoi model with increasing ice crystal size, such that it can be seen by readers how the model aggregation varies with size. What are also required in the revised paper are the Voronoi model's mass– and area–dimension power laws. These power law relations will go some way to explaining the single-scattering results and sensitivities of the Voronoi model to IWP and the characteristic size of the PSD in the brightness temperature difference sensitivity analysis. The fractal dimensions of mass and area of the Voronoi model are important in these respects.

Response: Thanks. The images of a set of seven Voronoi model shapes with increasing ice crystal size have been shown (Fig.3) in the literature of Ishimoto et al. (2012) as shown below. The geometrical characteristics of seven Voronoi model shapes can also be found in Table 1 of Ishimoto et al. (2012). Thus we cited the images and their geometrical characteristics of the Voronoi ice crystal models from Ishimoto et al. (2012) in section 2.1 as shown below. (See, Ishimoto, H., Masuda, K., Mano, Y., Orikasa, N., and Uchiyama, A.: Irregularly shaped ice aggregates in optical

modeling of convectively generated ice clouds, J Quant Spectrosc Ra, 113, 632-643, doi:10.1016/j.jqsrt.2012.01.017, 2012.)

Lines 145 - 147:

"As the particle size increases, the shape of the Voronoi ICS model changes and become complicated. The details of Voronoi model shapes with increasing ice crystal size have been shown and discussed in Ishimoto et al. (2012)".

[Figure]

Figure 2. Numerically created Voronoi aggregates for a model of irregular ice particles. See Fig. 3. cited from Ishimoto et al. (2012).

According to your comments, we have also added the Voronoi ice crystal model's mass- and area-dimension power laws in the revised manuscript. The mass-dimension and area-dimension power-law relationships of the Voronoi ice crystal model are defined by Ishimoto et al. (2012) and are described in Equation (1) - (3) as shown below.

$$m = 0.00528D^{2.1} \text{ (in cgs)} \tag{1}$$

$$A_r = 4G/\pi D^2 \tag{2}$$

$$A_r = 0.20D^{-0.29} \tag{3}$$

where the $m$ is the mass, $G$ is the cross-sectional area, $A_r$ is the area ratio and the $D$ is the maximum dimension of the Voronoi ice crystal model.

7. Apart from plotting the retrieved quantities, a further measure of how well the

Voronoi model represents the measured brightness temperatures at the three channels is to plot the residuals (i.e. brightness temperature differences between the forward model and measurements) as a function of time for all three channels.

Response: Thanks for the comments. The problem is the assumed water vapour, ozone and atmospheric profiles in the forward model can bring errors when converting the IWP and $D_{me}$ to the BTDs. So it is hard to explain the smallest brightness temperature differences between the forward model and measurements is due to the Voronoi model.

**Specific comments**

1. Introduction line 34. Since the authors discuss 20-30% of the global cloud mass, would it not be better to cite more updated studies that more directly measure the ice mass such as studies using CloudSat global retrievals of ice mass? As well as mm-wave retrievals of ice mass?

Response: According to the comments, we have added more studies relevant to ice mass retrievals using CloudSat and millimeter-wave data as shown below.

Lines 48-56: "The visible and infrared spectrum are sensitive to the visible optical depth and cloud top (Minnis et al., 1993b; Minnis et al., 1993a). The millimeter-wave ice cloud remote sensing technique is more suited to detect vertical cloud properties. Sensors such as the Millimeter-wave Imaging Radiometer (MIR) (Racette et al., 1992) and Special Sensor Microwave Water Vapor Sounder (SSM/T-2) have been used in several studies of IWP and particle size retrievals (Lin and Rossow, 1996; Liu and Curry, 1998, 1999). MIR channels at 89, 150 and 220 GHz have been used by Deeter and Evans (2000) and Liu and Curry (2000) to retrieve IWP and particle size in cirrus anvils over tropical ocean. Compared to passive sensors, the Cloud satellite radar (CloudSat), with an onboard millimeter-wavelength (94.05 GHz) radar and the raDAR/liDAR cloud product (DARDAR) (Ceccaldi et al., 2013) present new opportunities to infer the microphysical properties of ice clouds on a global scale. "

2. Line 36. As the paper is discussing Terahertz frequencies, another important property of large ice crystals that contribute to the radiative properties of ice cloud is their orientation.

Response: Thanks. We have added the "orientation" in this sentence as shown below.

Line 36: "Microphysical properties such as the ice water content, ice particle size, orientation, shape and etc. are the main influencing factors of the scattering and radiative properties of ice clouds."

3. List of citations on line 47. Fox (2020) should be added to this list?

Response: According to the comments, we have added the citation of the (Fox, 2020) in the list of citations on line 47 as shown below.

Lines 45-48: "Currently, large amounts of passive sensors (visible, infrared and microwave detectors) and related ice cloud retrieval algorithms (Nakajima and King, 1990; Nakajima et al., 1991, 2019; Nakajima and Nakajima, 1995; Platnick et al., 2003, 2017; Fox et al., 2019; Brath et al., 2018; Fox, 2020) have significantly developed."

4. Line 51. The description of Fox et al. (2019) needs to be more accurate, the study also used sub-mm frequencies up to 664 GHz, and in Fox (2020). The works of Fox (2019,2020) includes the Terahertz region, and not just the microwave.

Response: Thanks for the comments, we have modified the irrational expressions on lines 50-51 as follows: "Additionally, microwave regions are mainly useful for large particles larger than 500 μm compared to the terahertz region (Fox, 2020; Fox et al., 2019).".

5. Line 79, again ice crystal orientation is also an important consideration here.

Response: We have rewritten this sentence as follows: "Different assumptions of ice cloud microphysical properties (shape, size, orientation and particle size distribution of ice particles) in the forward physical model significantly affect the retrieval of the IWP and particle size of ice clouds.".

6. Line 97. Another numerical method that could be included in this list is the Boundary Element Method, which has recently been applied to very complex ice crystals by Kleanthous et al. (2022): Antigoni Kleanthous, Timo Betcke, David P. Hewett, Paul Escapil-Inchauspé, Carlos Jerez-Hanckes, Anthony J. Baran, Accelerated Calderón preconditioning for Maxwell transmission problems, Journal of Computational Physics, Volume 458, 2022, 111099, ISSN 0021-9991,

https://doi.org/10.1016/j.jcp.2022.111099. A further paper here could be Mano (2000), who applied BEM to hexagonal ice columns. "Exact solution of electromagnetic scattering by a three-dimensional hexagonal ice column obtained with the boundary-element method," Appl. Opt. 39, 5541-5546.

Response: According to the comments, we have included the Boundary Element Method in the introduction section (Lines 97-99) as follows: "Moreover, the boundary element method (Groth et al., 2015; Kleanthous et al., 2022) has been recently applied to complex ice particles.".

7. Line 98. This GOA acronym has not been defined - should it be GOM?

Response: Thanks. We have added the definition of the "GOA" as shown below.

Line 98: ".. several improved Geometrical Optics Approximation (GOA) methods .."

8. The discussion beginning on line 108. Another ICS model worthy of note in this context is the ensemble model of cirrus ice crystals developed by Baran and Labonnote (2007). The ensemble model attempts to be more representative of the evolution of the ice crystal aggregation process as a function of increasing size, see Baran, A.J. and Labonnote, L.-C. (2007), A self-consistent scattering model for cirrus. I: The solar region. Q.J.R. Meteorol. Soc., 133: 1899-1912. ) https://doi.org/10.1002/qj.164), and Baran et al. 2014 (Baran, A.J., Cotton, R., Furtado, K., Havemann, S., Labonnote, L.-C., Marenco, F., Smith, A. and Thelen, J.-C. (2014), A self-consistent scattering model for cirrus. II: The high and low frequencies. Q.J.R. Meteorol. Soc., 140: 1039-1057. https://doi.org/10.1002/qj.2193).

Response: According to the comments, we have included the ensemble ice crystal model in the introduction section (Lines 108-110) and added corresponding descriptions as follows: "Furthermore, features including various habit ensembles were added into ice particles. For example, Baran and Labonnote (2007) and Baran et al. (2014a) developed an ensemble ice particle model made of hexagonal column ice particles for use in the Met Office Unified Model Global Atmosphere 5.0 (GA5) configuration (Baran et al., 2014b).".

9. Typo on line 117 Mo.,del -> Model

Response: We have corrected this error on line 117 as shown below.

Line 117: "the Community Integrated Earth System Model (CIESM)."

删除[liming]: Mo.,del

10. Line 118. The word effectiveness is sufficient, using the word "superiority" is inappropriate here because it has not been proven relative to all other models that are now available.

Response: According to the comments, we have removed the word "superiority" on line 118.

11. Line 122. ICI is not correct here, the instrument is ISMAR (International Sub-Millimeter wave Airborne Radiometer) described in Fox et al., 2017. ISMAR was jointly funded by the Met Office and ESA - not ICI.

Response: We have corrected this error on line 122 as shown below.

Line 122: "The database of the Voronoi ICS model in the terahertz region was adopted by Baran et al. (2018) as standard data for the modelling and evaluation of the ISMAR (International Sub-Millimeter wave Airborne Radiometer) which the European Space Agency (ESA) and the Met Office jointly developed (Kangas et al., 2014; Fox et al., 2017)."

12. Section 2.1. Which refractive indices are being used to compute the SSPs? The refractive indices in the microwave and sub-millimeter are temperature dependent - is this dependence considered in the simulations that follow? If not, which temperature has been assumed in the calculation of the SSPs? How have you justified the selection of this temperature?

Response: For the first question, according to the suggestions, we have added specifications and the reference on lines 148-149 as follows: "The calculation of the single-scattering property utilize the real and imaginary parts of ice from the newest library of the refractive index provided by Warren and Brandt (2008).".

For the second question, we utilized the refractive indices of ice in the microwave and sub-millimeter at fixed temperature. In this study, the refractive indices of the Voronoi ICS model at the frequencies of 10-874 GHz are computed at temperature 266 K according to Warren and Brandt (2008). And the refractive indices of the Column ICS model added in this study at the frequencies of 89-340 GHz are derived from Warren (1984) at the temperature of -30°C.

For the third question, according to Warren and Brandt (2008), the updated refractive indices were shown only for a single temperature, 266 K. Furthermore, Kim (2006) has indicated that the dependence of refractive indexes of ice crystals on temperature causes only 1% difference in the SSPs at the microwave frequencies. Several studies (Hong, 2007; Hong et al., 2009) have also utilized the refractive indices of ice crystals at assumed temperature. Hence, in this study, the influence of the refractive indices of ice at the fixed temperature on the calculations of the SSPs is ignored here.

13. Section 2.3. Is the cloud between the boundaries assumed to be homogeneous? If so, please state this.

Response: According to the suggestions, we have added the statement of this assumption on lines 168-169 as follows: "The RSTAR radiative transfer model assumes the simulated scene is composed of a homogeneous ice cloud layer".

14. Equation 2, line 186. In the denominator, this is why you need to provide the model's mass - dimension relationship.

Response: According to the comments, we have added the mass-dimension power-law relationship of the Voronoi ice crystal model as shown below.

$$m = 0.00528D^{2.1} \ \text{(in cgs)} \tag{1}$$

where $m$ is the mass and $D$ is the maximum dimension of the Voronoi ice crystal model."

15. Equation 4, line 205. In the denominator, this is why you need to provide the model's area - dimension relationship.

Response: We have added the area-dimension power-law relationship of the Voronoi ice crystal model as shown below.

$$A_r = 4G/\pi D^2 \tag{1}$$

$$A_r = 0.20D^{-0.29} \tag{2}$$

where $G$ is the cross-sectional area, $A_r$ is the area ratio and $D$ is the maximum dimension of the Voronoi ice crystal model.

16. Equation 7, there is a missing wavelength dependence in the denominator for the scattering cross section.

Response: We have corrected this error in Equation 7 as shown below.

$$g(\lambda) = \frac{\int_{L_{min}}^{L_{max}} g(\lambda,L)\sigma_{sca}(\lambda,L)n(L)dL}{\int_{L_{min}}^{L_{max}} \sigma_{sca}(\lambda,L)n(L)dL} \quad , \tag{7}$$

17. Equations 9 -12, how accurate are the parametric fits as a function of De?

Response: For the equations (9)-(12), the mean single-scattering properties are functions of wavelengths and depend on the particle size distribution. There are diverse ways to define the effective particle size of nonspherical ice crystals in the literature. Following Pollack and Cuzzi (1980), Foot (1988) and Mitchell (2002), for irregularly-shaped large particles the absorption coefficient depends on the volume of the particle, and the scattering coefficient depends on the cross-sectional area. Hence, the effective particle size of nonspherical particles associated with a given size distribution is defined as follows:

$$D_e = \frac{3}{2}\frac{\int_{L_{min}}^{L_{max}} V(L)n(L)dL}{\int_{L_{max}}^{L_{max}} A(L)n(L)dL} \quad , \tag{4}$$

where $D_e$ is the effective particle diameter, V and A are the volume and projected area of Voronoi and Sphere models. The effective particle diameter $D_e$ can account for the shape of irregular ice crystals and provide a measure of the average size of the cloud particles for a given size distribution. Currently, the parametric fits as a function of $D_e$ has been used in several general parameterization schemes (Baum et al., 2005a; Baum et al., 2005b; Fu, 1996; Mitchell et al., 1996b; Mitchell et al., 2006; Yi et al., 2013).

18. Figure 9, this figure might be better plotted as a PDF of the retrievals, and statistically measure how different the distributions are from the reference PDFs using some statistical measure.

Response: According to the suggestions, we have modified the scatter plot in Figure 9 to a PDF plot, as shown below.

[Figure]

Figure 9: The joint histogram of differences of (a) the IWP and (b) Dme between the retrieved results and the CoSSIR-MCBI algorithm results for the Voronoi (red line), Sphere (black line) and Column models (blue line), separately.

Reference

Baran, A., Ishimoto, H., Sourdeval, O., Hesse, E., and Harlow, C.: The applicability of physical optics in the millimetre and sub-millimetre spectral region. Part II: Application to a three-component model of ice cloud and its evaluation against the bulk single-scattering properties of various other aggregate models, Journal of Quantitative Spectroscopy and Radiative Transfer, 206, doi:10.1016/j.jqsrt.2017.10.027, 2018.

Baran, A. J. and Labonnote, L. C.: A self-consistent scattering model for cirrus. I: The solar region, Q J Roy Meteor Soc, 133, 1899-1912, 10.1002/qj.164, 2007.

Baran, A. J., Hill, P., Furtado, K., Field, P., and Manners, J.: A Coupled Cloud Physics Radiation Parameterization of the Bulk Optical Properties of Cirrus and Its Impact on the Met Office Unified Model Global Atmosphere 5.0 Configuration, J Climate, 27, 7725-7752, 2014b.

Baran, A. J., Cotton, R., Furtado, K., Havemann, S., Labonnote, L. C., Marenco, F., Smith, A., and Thelen, J. C.: A self-consistent scattering model for cirrus. II: The high and low frequencies, Q J Roy Meteor Soc, 140, 1039-1057, 10.1002/qj.2193, 2014a.

Baum, B. A., Heymsfield, A. J., Yang, P., and Bedka, S. T.: Bulk scattering properties for the remote sensing of ice clouds. Part I: Microphysical data and models, J Appl Meteorol, 44, 1885-1895, doi:10.1175/JAM2308.1, 2005a.

Baum, B. A., Yang, P., Heymsfield, A. J., Platnick, S., King, M. D., Hu, Y. X., and Bedka, S. T.: Bulk scattering properties for the remote sensing of ice clouds. Part II: Narrowband models, J Appl Meteorol, 44, 1896-1911, doi:10.1175/JAM2309.1, 2005b.

Brath, M., Fox, S., Eriksson, P., Harlow, C., Burgdorf, M., and Buehler, S.: Retrieval of an ice water path over the ocean from ISMAR and MARSS millimeter and

submillimeter brightness temperatures, Atmospheric Measurement Techniques, 11, 611-632, doi:10.5194/amt-11-611-2018, 2018.

Ceccaldi, M., Delanoë, J., Hogan, R., Pounder, N., Protat, A., and Pelon, J.: From CloudSat-CALIPSO to EarthCare: Evolution of the DARDAR cloud classification and its comparison to airborne radar-lidar observations, Journal of Geophysical Research: Atmospheres, 118, 10.1002/jgrd.50579, 2013.

Deeter, M. and Evans, K.: A Novel Ice-Cloud Retrieval Algorithm Based on the Millimeter-Wave Imaging Radiometer (MIR) 150- and 220GHz Channels, Journal of Applied Meteorology - J APPL METEOROL, 39, 623-633, 10.1175/1520-0450-39.5.623, 2000.

Foot, J.: Some observations of the optical properties of clouds. II: Cirrus, Q J Roy Meteor Soc, 114, 145-164, 10.1002/qj.49711447908, 1988.

Fox, S.: An Evaluation of Radiative Transfer Simulations of Cloudy Scenes from a Numerical Weather Prediction Model at Sub-Millimetre Frequencies Using Airborne Observations, Remote Sensing, 12, 2758, 10.3390/rs12172758, 2020.

Fox, S., Mendrok, J., Eriksson, P., Ekelund, R., amp, apos, Shea, S., Bower, K., Baran, A., Harlow, C., and Pickering, J.: Airborne validation of radiative transfer modelling of ice clouds at millimetre and sub-millimetre wavelengths, Atmospheric Measurement Techniques, 12, 1599-1617, doi:10.5194/amt-12-1599-2019, 2019.

Fox, S., Lee, C., Moyna, B., Philipp, M., Rule, I., Rogers, S., King, R., Oldfield, M., Rea, S., Henry, M., Wang, H., and Harlow, C.: ISMAR: An airborne submillimetre radiometer, Atmospheric Measurement Techniques, 10, 477-490, doi:10.5194/amt-10-477-2017, 2017.

Fu, Q. A.: An accurate parameterization of the solar radiative properties of cirrus clouds for climate models, J Climate, 9, 2058-2082, 1996.

Graham, R., Hudson, S., and Maturilli, M.: Improved Performance of ERA5 in Arctic Gateway Relative to Four Global Atmospheric Reanalyses, Geophysical Research Letters, 46, 10.1029/2019GL082781, 2019.

Groth, S. P., Baran, A. J., Betcke, T., Havemann, S., and Smigaj, W.: The boundary element method for light scattering by ice crystals and its implementation in BEM plus, J Quant Spectrosc Ra, 167, 40-52, 2015.

Heymsfield, A. J., Schmitt, C., and Bansemer, A.: Ice Cloud Particle Size Distributions and Pressure-Dependent Terminal Velocities from In Situ Observations at Temperatures from 0 degrees to-86 degrees C, J Atmos Sci, 70, 4123-4154, 2013.

Hong, G.: Parameterization of scattering and absorption properties of nonspherical ice crystals at microwave frequencies, Journal of Geophysical Research, 112, 10.1029/2006JD008364, 2007.

Hong, G., Yang, P., Baum, B. A., Heymsfield, A. J., Weng, F., Liu, Q., Heygster, G., and Buehler, S. A.: Scattering database in the millimeter and submillimeter wave range of 100–1000 GHz for nonspherical ice particles, Journal of Geophysical Research: Atmospheres, 114, doi:10.1029/2008JD010451, 2009.

Ishimoto, H., Masuda, K., Mano, Y., Orikasa, N., and Uchiyama, A.: Irregularly

shaped ice aggregates in optical modeling of convectively generated ice clouds, J Quant Spectrosc Ra, 113, 632-643, doi:10.1016/j.jqsrt.2012.01.017, 2012.

Kangas, V., D'Addio, S., Klein, U., Loiselet, M., Mason, G., Orlhac, J.-C., Gonzalez, R., Bergada, M., Brandt, M., and Thomas, B.: Ice cloud imager instrument for MetOp Second Generation, 2014 13th Specialist Meeting on Microwave Radiometry and Remote Sensing of the Environment(MicroRad), 228-231, doi:10.1109/MicroRad.2014.6878946, 2014.

Kim, M.-J.: Single scattering parameters of randomly oriented snow particles at microwave frequencies, J. Geophys. Res, 111, 10.1029/2005JD006892, 2006.

Kleanthous, A., Betcke, T., Hewett, D., Escapil-Inchauspé, P., Jerez-Hanckes, C., and Baran, A.: Accelerated Calderón preconditioning for Maxwell transmission problems, Journal of Computational Physics, 458, 111099, 10.1016/j.jcp.2022.111099, 2022.

Li, M., Letu, H., Peng, Y., Ishimoto, H., Lin, Y., Nakajima, T. Y., Baran, A. J., Guo, Z., Lei, Y., and Shi, J.: Investigation of ice cloud modeling capabilities for the irregularly shaped Voronoi ice scattering models in climate simulations, Atmos. Chem. Phys., 22, 4809-4825, doi:10.5194/acp-22-4809-2022, 2022.

Lin, B. and Rossow, W.: Seasonal Variation of Liquid and Ice Water Path in Nonprecipitating Clouds over Oceans, Journal of Climate - J CLIMATE, 9, 2890-2902, 10.1175/1520-0442(1996)009<2890:SVOLAI>2.0.CO;2, 1996.

Liu, G. and Curry, J.: Remote Sensing of Ice Water Characteristics in Tropical Clouds Using Aircraft Microwave Measurements, Journal of Applied Meteorology - J APPL METEOROL, 37, 337-355, 10.1175/1520-0450(1998)037<0337:RSOIWC>2.0.CO;2, 1998.

Liu, G. and Curry, J.: Tropical Ice Water Amount and Its Relations to Other Atmospheric Hydrological Parameters as Inferred from Satellite Data, Journal of Applied Meteorology - J APPL METEOROL, 38, 1182-1194, 10.1175/1520-0450(1999)038<1182:TIWAAI>2.0.CO;2, 1999.

Liu, G. and Curry, J.: Determination of Ice Water Path and Mass Median Particle Size Using Multichannel Microwave Measurements, Journal of Applied Meteorology - J APPL METEOROL, 39, 1318-1329, 10.1175/1520-0450(2000)039<1318:DOIWPA>2.0.CO;2, 2000.

Minnis, P., Heck, P., and Young, D.: Inference of Cirrus Cloud Properties Using Satellite-observed Visible and Infrared Radiances. Part II: Verification of Theoretical Cirrus Radiative Properties, J Atmos Sci, 50, 10.1175/1520-0469(1993)050<1305:IOCCPU>2.0.CO;2, 1993b.

Minnis, P., Liou, K.-N., and Takano, Y.: Inference of Cirrus Cloud Properties Using Satellite-observed Visible and Infrared Radiances. Part I: Parameterization of Radiance Fields, J Atmos Sci, 50, 10.1175/1520-0469(1993)050<1279:IOCCPU>2.0.CO;2, 1993a.

Mitchell, David, L., Liu, Yangang, Macke, and Andreas: Modeling Cirrus Clouds. Part II: Treatment of Radiative Properties, J. Atmos. Sci, 53, 2967-2988, 1996b.

Mitchell, D.: Effective Diameter in Radiation Transfer: General Definition, Applications, and Limitations, Journal of The Atmospheric Sciences - J ATMOS

SCI, 59, 2330-2346, 10.1175/1520-0469(2002)059<2330:EDIRTG>2.0.CO;2, 2002.

Mitchell, D. L., Baran, A. J., Arnott, W. P., and Schmitt, C.: Testing and comparing the modified anomalous diffraction approximation, J Atmos Sci, 63, 2948-2962, 2006.

Nakajima, T. and King, M. D.: Determination of the Optical-Thickness and Effective Particle Radius of Clouds from Reflected Solar-Radiation Measurements .1. Theory, J Atmos Sci, 47, 1878-1893, doi:10.1175/1520-0469(1990)047<1878:Dotota>2.0.Co;2, 1990.

Nakajima, T., King, M. D., Spinhirne, J. D., and Radke, L. F.: Determination of the Optical-Thickness and Effective Particle Radius of Clouds from Reflected Solar-Radiation Measurements .2. Marine Stratocumulus Observations, J Atmos Sci, 48, 728-750, doi:10.1175/1520-0469(1991)048<0728:Dotota>2.0.Co;2, 1991.

Nakajima, T. Y. and Nakajima, T.: Wide-area determination of cloud microphysical properties from NOAA AVHRR measurements for FIRE and ASTEX regions, J Atmos Sci, 52, 4043-4059, doi:10.1175/1520-0469(1995)052<4043:Wadocm>2.0.Co;2, 1995.

Nakajima, T. Y., Ishida, H., Nagao, T. M., Hori, M., Letu, H., Higuchi, R., Tamaru, N., Imoto, N., and Yamazaki, A.: Theoretical basis of the algorithms and early phase results of the GCOM-C (Shikisai) SGLI cloud products, Prog Earth Planet Sc, 6, doi:10.1186/s40645-019-0295-9, 2019.

Platnick, S., King, M. D., Ackerman, S. A., Menzel, W. P., Baum, B. A., Riedi, J. C., and Frey, R. A.: The MODIS cloud products: algorithms and examples from Terra, IEEE Transactions on Geoscience & Remote Sensing, 41, 459-473, doi:10.1109/TGRS.2002.808301, 2003.

Platnick, S., Meyer, K. G., King, M. D., Wind, G., Amarasinghe, N., Marchant, B., Arnold, G. T., Zhang, Z. B., Hubanks, P. A., Holz, R. E., Yang, P., Ridgway, W. L., and Riedi, J.: The MODIS Cloud Optical and Microphysical Products: Collection 6 Updates and Examples From Terra and Aqua, Ieee T Geosci Remote, 55, 502-525, doi:10.1109/Tgrs.2016.2610522, 2017.

Pollack, J. and Cuzzi, J.: Scattering by Nonspherical Particles of Size Comparable to a Wavelength: A New Semi-Empirical Theory and Its Application to Tropospheric Aerosols, J Atmos Sci, 37, 10.1175/1520-0469(1980)037<0868:SBNPOS>2.0.CO;2, 1980.

Racette, P., Dod, L., Shiue, J., Adler, R., Jackson, D., Gasiewski, A., and Zacharias, D.: An Airborne Millimeter-Wave Imaging Radiometer for Cloud, Precipitation, and Atmospheric Water Vapor Studies, Journal of Atmospheric and Oceanic Technology, 13, 10.1175/1520-0426(1996)013<0610:AAMWIR>2.0.CO;2, 1992.

Warren, S.: Optical Constants of Ice from the Ultraviolet to the Microwave, Applied optics, 23, 1206, 10.1364/AO.23.001206, 1984.

Warren, S. G. and Brandt, R. E.: Optical constants of ice from the ultraviolet to the microwave: A revised compilation, J Geophys Res-Atmos, 113, 2008.

Yang, P., Liou, K. N., Wyser, K., and Mitchell, D.: Parameterization of the scattering and absorption properties of individual ice crystals, J Geophys Res-Atmos, 105, 4699-4718, 2000a.

Yi, B. Q., Yang, P., Baum, B. A., L'Ecuyer, T., Oreopoulos, L., Mlawer, E. J., Heymsfield, A. J., and Liou, K. N.: Influence of Ice Particle Surface Roughening on the Global Cloud Radiative Effect, J Atmos Sci, 70, 2794-2807, 2013.

---

## Author Comment (AC3)

Comment on amt-2022-247

Anonymous Referee # 4

Referee comment RC3 on "Retrieval of Terahertz Ice Cloud Properties from airborne measurements based on the irregularly shaped Voronoi ice scattering models" by Ming Li et al.

**General comments**

This study implements Voronoi and spherical ice crystal models in observed brightness temperatures at 380, 640 and 874 GHz to retrieve Ice water path(IWP) and effective particle radius (Re). Authors show that Voronoi model can better reproduce results compared to previous 'standard' values. I think such result is obvious and can be easily predicated as simple spherical model is less adequate for irregular-shaped ice particles, but authors showed it quantitatively. In my opinion, there exist mistakes/incompleteness in English language, methodology, and scientific discussions, as outlined in specific comments below. Authors are suggested to improve the manuscript by considering the comments given below as well as by carefully improving the English writing/expression.

Response: Thank you very much for your significant comments.

**Specific comments**

1. Line 19: '.. we completed the..'--> completed is what sense? rewrite it.

Response: We have rewritten it as follows: ".. we developed the Voronoi ..".

2. Line 48 : '..signfinifiantly developed...': rewrite the sentence.

Response: We have rewritten this sentence as follows: "Currently, large amounts of passive sensors (visible, infrared and microwave detectors) have been developed and related ice cloud retrieval algorithms have been reported in substantial literature (Nakajima and King, 1990; Nakajima et al., 1991, 2019; Nakajima and Nakajima, 1995; Platnick et al., 2003, 2017; Fox et al., 2019; Brath et al., 2018).".

3. Line 132: What is 'CoSSIR-MCBI' algorithm? A brief description is important with relevant references.

Response: According to the comments, we have illustrated the definition of the "CoSSIR-MCBI" abbreviation on lines 73 - 76 as follows: "Evans et al. (2002) developed a Monte Carlo Bayesian Integration (MCBI) algorithm to retrieve ice clouds' IWP and median mass diameter ($D_{me}$) from simulated SWCIR brightness temperatures. Then, Evans et al. (2005) applied the MCBI method to retrieve IWP and $D_{me}$ using the CoSSIR brightness temperatures (referred to as the CoSSIR-MCBI hereafter).".

4. Lines 140-143: Specify ice particle sizes either in table or describe size interval in the text. Provide same information for the 20 wavelengths mentioned here.

Response: According to the comments, we have added one table (Table 1) to list the 31 ice particle sizes and 20 wavelengths contained in the single-scattering property database of the Voronoi ICS model. Table 1 is shown below. We have also modified the corresponding descriptions on lines 140-142 as follows: "For the Voronoi ICS model, the single-scattering property database contains 31 ice particle sizes ranging from 0.25 to 9300 μm and covers 20 terahertz channels with frequencies ranging from 10 to 874 GHz, corresponding to wavelengths from 0.03 to 3cm (see Table 1).".

Table 1. The 20 frequency channels and 31 maximum dimensions of ice particles included in the single-scattering property database of the Voronoi ICS model.

| Maximum dimension (μm) | Frequency (GHz) | Wavelength (cm) |
|---|---|---|
| 0.400E+00 | 0.10000E+02 | 0.29979E+01 |
| 0.100E+01 | 0.15000E+02 | 0.19986E+01 |
| 0.200E+01 | 0.18700E+02 | 0.16032E+01 |
| 0.300E+01 | 0.23800E+02 | 0.12596E+01 |
| 0.500E+01 | 0.31400E+02 | 0.95475E+00 |
| 0.750E+01 | 0.35000E+02 | 0.85655E+00 |
| 0.150E+02 | 0.50300E+02 | 0.59601E+00 |
| 0.250E+02 | 0.53750E+02 | 0.55775E+00 |
| 0.350E+02 | 0.55000E+02 | 0.54508E+00 |
| 0.450E+02 | 0.89000E+02 | 0.33685E+00 |

| | | |
|---|---|---|
| 0.600E+02 | 0.94000E+02 | 0.31893E+00 |
| 0.700E+02 | 0.11875E+03 | 0.25246E+00 |
| 0.147E+03 | 0.16550E+03 | 0.18114E+00 |
| 0.225E+03 | 0.18331E+03 | 0.16354E+00 |
| 0.314E+03 | 0.22900E+03 | 0.13091E+00 |
| 0.419E+03 | 0.24300E+03 | 0.12337E+00 |
| 0.500E+03 | 0.32500E+03 | 0.92244E-01 |
| 0.623E+03 | 0.44800E+03 | 0.66918E-01 |
| 0.752E+03 | 0.66400E+03 | 0.45150E-01 |
| 0.867E+03 | 0.87400E+03 | 0.34301E-01 |
| 0.964E+03 | | |
| 0.108E+04 | | |
| 0.140E+04 | | |
| 0.175E+04 | | |
| 0.256E+04 | | |
| 0.350E+04 | | |
| 0.500E+04 | | |
| 0.750E+04 | | |
| 0.100E+05 | | |
| 0.120E+05 | | |
| 0.150E+05 | | |

5. Line 149: Specify what refractive index values are used for ice particles here. Give a reference.

Response: According to the comments, we have added specifications and the reference on lines 148-149 as follows: "Calculations of the single-scattering property utilize the real and imaginary parts of ice from the newest library of the refractive index provided by Warren and Brandt (2008).".

6. I think Figure 1 is confusing. Either make Figure 1 clear or remove it and describe the mythology clearly in the text.

Response: According to the comments, we have redrawn the Figure 1 as shown below. We updated Figure 1 with the inclusion of the hexagonal column ice crystal scattering model from the ARTS collection of models to compare with the Voronoi model results.

[Figure]

Figure 1: The overall flowchart of the retrieval of the IWP and $D_{me}$ of ice clouds based on the Voronoi, Sphere and Column ICS models.

7. Line 179: I do not understand how clear sky days are selected here. Are they before and/or after the cloudy sky days or average of certain week (or month etc.)?

Response: Here we simulated a clear-sky observed scenario based on the radiative transfer model. We are aimed to construct a clear-sky look-up table with different inputs of water vapour and ozone columns. This clear-sky look-up table is used later in the retrieval process that follows.

8. Lines 188-189: 'statistical multiple linear regression method': Write a few lines to clarify it. For example, what are dependent and independent variables in this method?

Response: According to the comments, we have added more illustrations of the "statistical multiple linear regression method" on lines 188-191 as follows: "Due to

the different definitions of $R_e$ and $D_{me}$, a transformation of the particle size descriptors is necessary. A statistical multiple linear regression was used in the transformation. Firstly, the ice particle number concentration of ice clouds was specified. A total of 14,408 groups of PSDs from aircraft observation sampling data were selected. The equivalent-volume sphere's particle radius, maximum particle dimension and mass were used to integrate over 14,408 PSDs. Then 14,408 groups of $R_e$ and $D_{me}$ were implemented. Finally, we build a relationship between the $R_e$ and $D_{me}$ and coefficients can be obtained by numerical fitting and provided as input. We regard the $R_e$ and $D_{me}$ as independent and dependent variables, respectively. Therefore, $D_{me}$ can be calculated from the coefficients.".

9. Line 199: A reference for Eq. (3) is required.

Response: According to the comments, we have added a reference for Eq. (3) as follows: ".. we adopt the gamma distribution form following Heymsfield et al. (2013) as follows ..".

10. Lines214-219: Please specify the coefficient terms either in table or in text.

Response: According to the comments, we have added four tables (Table A.1-A.4) listing the coefficient terms of Eq. (9)-(12) as shown below. We have added descriptions as follows: "Values of the above coefficients for Voronoi scheme are listed in appendix A (Tables A.1, A.2, A.3, and A.4).".

Table A.1

Coefficients in the fitting of terahertz mass extinction coefficients ($m^2$ /g).

| Frequency (GHz) | $a_0$ ($m^2$/g) | $a_1$ ($m^3$/g) |
| --- | --- | --- |
| 325 | 7.0891e-01 | -1.6965e+01 |
| 448 | 2.1347e+00 | -5.0405e+01 |
| 664 | 7.5009e+00 | -1.6770e+02 |
| 874 | 1.5790e+01 | -3.2850e+02 |

Table A.2

Coefficients in the fitting of terahertz single-scattering albedo.

| Frequency (GHz) | $b_0$ | $b_1$ | $b_2$ | $b_3$ |
|---|---|---|---|---|
| 325 | -3.1317e-01 | 2.7448e-02 | -2.0449e-04 | 5.0815e-07 |
| 448 | -2.3947e-01 | 2.9461e-02 | -2.4145e-04 | 6.4366e-07 |
| 664 | -8.2857e-02 | 2.7985e-02 | -2.4357e-04 | 6.7691e-07 |
| 874 | 4.7425e-02 | 2.5164e-02 | -2.2395e-04 | 6.3152e-07 |

Table A.3

Coefficients in the fitting of terahertz asymmetry factor.

| Frequency (GHz) | $c_0$ | $c_1$ | $c_2$ | $c_3$ |
|---|---|---|---|---|
| 325 | 2.2045e-02 | -8.2487e-04 | 2.5764e-05 | -4.7767e-08 |
| 448 | 1.0168e-02 | -5.1223e-05 | 3.0599e-05 | -8.0591e-08 |
| 664 | -4.4704e-02 | 3.5331e-03 | 1.2997e-05 | -7.2297e-08 |
| 874 | -1.1685e-01 | 8.8403e-03 | -3.0410e-05 | 2.6790e-08 |

Table A.4

Coefficients in the fitting of terahertz mass-averaged absorption coefficients ($m^2$ /g).

| Frequency (GHz) | $d_0$ ($m^2$/g) | $d_1$ (m/g) | $d_2$ (1/g) | $d_3$ ($m^{-1}$/g) |
|---|---|---|---|---|
| 325 | 4.4262e-02 | 1.5585e-04 | 9.6647e-07 | -5.1271e-09 |
| 448 | 8.2110e-02 | 5.0544e-04 | 2.0336e-06 | -1.2945e-08 |
| 664 | 1.6909e-01 | 2.4299e-03 | 1.2784e-06 | -3.3930e-08 |
| 874 | 2.6509e-01 | 7.6295e-03 | -1.4488e-05 | -3.9275e-08 |

11. L222: Is $BTD_{1-3}$ is same to $BTD_{1-2}$-$BTD_{2-3}$ ? If so, better to write $BTD_{1-3}$ instead of $BTD_{1-2}$-$BTD_{2-3}$.

Response: According to the comments, we have replaced the $BTD_{1-2}$-$BTD_{2-3}$ with $BTD_{1-3}$ and have unified them in the revised manuscript. The relevant descriptions are shown below.

Lines 221-223: "The difference between the 640 GHz BTD and the 874 GHz BTD is simplified to $BTD_{2-3}$. And the difference between the 380 GHz BTD and the 640 GHz

BTD is simplified to $BTD_{1-2}$. We named the difference between the $BTD_{1-2}$ and $BTD_{2-3}$ as $BTD_{1-3}$."

12. Lines 227-229: I think BTD depends strongly on cloud top temperature as well as surface temperature along with cloud properties. Since they are fixed here, errors are expected in retrieved values. Can authors provide error ranges in retrieved parameters due to such assumptions? If possible, authors are suggested to use actual data from cloud top and surface temperatures rather than assumptions.

Response: Thanks for the comments. For large particles (SZP>1), the scattering is mainly Mie scattering, and the single scattering albedo is close to 1 (Figure 1). Hence, ice particle scattering plays a leading role. Since the radiation is mainly scattered by ice crystal particles, the absorption effects and emission effects are small, and the BTD THz radiation at the top of the atmosphere is almost independent of cloud top temperature.

For the error caused by the surface temperature assumptions, the clear-sky atmospheric optical thickness for the 0.3-3 Thz is large as shown in Figure 2. Thus most of the radiation emitted by the surface temperature is absorbed by the lower layer of water vapor and ozone. Furthermore, there is a lack of actual surface temperature data synchronized with the airborne measurement track. Furthermore, the reanalysis data such as ERA5 with coarse spatial resolution will introduce new errors to retrieval results.

[Figure]

Figure 1: The extinction efficiency, single-scattering albedo and asymmetry factor as functions of the SZP for the Voronoi (blue solid line), Sphere (red dashed line) and Column (green dashed line) ICS models in the (a, c, e) 325 and (b, d, f) 874 GHz frequencies.

[Figure]

Figure 2. The spectral variations of the clear-sky atmospheric optical thickness under four atmospheric conditions.

13. A theoretical perspective should be given for using $BTD_{2-3}$ and $BTD_{1-2} - BTD_{2-3}$ in Eq. 4. (Why these two are important in cloud properties retrieval among several possible combinations of three wavelengths). Further, there should be an error term in Eq. 13. It is further necessary to describe in detail about the methodology. For example, what are the convergence criteria, how the initial values are determined, and

how measurement errors can affect the retrieved values etc.

Response: Thanks for the comments. For the first question, the basis of using the combination of the $BTD_{2-3}$ and $BTD_{1-2}$ -$BTD_{2-3}$ is as follows: The 380 GHz is the atmospheric absorption peak, while 640 and 874 GHz are the atmospheric windows as shown in Figure 2 below. Therefore, both the 640 and 874 GHz brightness temperature are affected by ice clouds, while the brightness temperature of 380 GHz is insensitive to ice cloud microphysical properties. Hence, the 380 minus 640 GHz brightness temperature differences ($BTD_{1-2}$) can highlight the brightness temperature depression caused by ice clouds. And the 640 minus 874 GHz brightness temperature differences ($BTD_{2-3}$) can reflect the difference in the scattering properties of differently shaped ice clouds. This is helpful to study the role of different ice crystal shapes. The differences between 640 and 874 GHz also can offset the regional errors due to different latitudes, atmospheric profiles and atmospheric states. In summary, The $BTD_{2-3}$ and $BTD_{1-2}$ -$BTD_{2-3}$ combination can integrate the information of three frequencies and show the sensitivity to ice clouds, eliminating the impacts of different atmospheric profiles and conditions.

[Figure]

Figure 2. The spectral variations of the clear-sky atmospheric optical thickness under four atmospheric conditions.

For the second question, we have modified the Eq. (13)-(14) and added the detailed method as shown below.

"

$$Y = F(X) + \epsilon \, , \tag{13}$$

$$X = \begin{pmatrix} IWP, \\ D_{me} \end{pmatrix} \, , \; F(X) = \begin{pmatrix} BTD_{2-3}, \\ BTD_{1-2} - BTD_{2-3} \end{pmatrix}, \tag{14}$$

where $X$ is the vector-matrix composed of the variables of the $IWP$ and $D_{me}$ to be solved. $Y$ is the vector composed of the two BTDs and the uncertainty vector. The vector $\epsilon$ represents the uncertainties that are attached to the measurements (i.e. instrumental accuracy) and to the radiative transfer forward model (i.e. approximation errors in the radiative transfer model). Following Marks and Rodgers (1993), a good convergence can be obtained when the value of the cost function is lower than the size of the measurement vector. Since there is no robust a priori for $IWP$ and $D_{me}$, we selected an average value as the initial value for a priori value of the $IWP$ and $D_{me}$."

14. Line 250: Since the x-axis is size parameter (not radius), it is difficult to understand where 120um exist. Either rewrite the text or make Figure 2 clear by adding additional x-axis.

Response: According to the comments, we have rewritten the text as follows: "For small ice particles with SZPs less than 1, the single-scattering properties are small and barely influenced by the shape of ice particles.".

15. Lines 250-254: Why large difference exists for large sized particles in Figure 2 remains undiscussed.

Response: Thanks for the comments, we have added discussions on lines 254-256 as follows: "As ice particle size increases, scattering is predominantly Mie scattering and sensitivity of the single-scattering properties to the ice crystal habits becomes pronounced, so that ice crystal shape contributes to the large differences for large particle sizes. ".

16. Lines 261-264: What could be the plausible reasons for such results for relatively larger particles?

Response: We have added reasons on lines 261-264 as shown below.

Lines 273-277: "On the one hand, the higher extinction efficiency and single-scattering albedo of the Voronoi ICS model for large particles are possibly due to the multifaceted shapes of the Voronoi ICS model, which can result in significant

side and backward scattering and increase the scattered energy. On the other hand, for large particles, the higher asymmetry factor of the Voronoi ICS model is possibly because the scattered energy is dominated by diffraction. The diffracted energy is concentrated in the forward direction, leading to a large asymmetry factor."

17. I do not understand why BTD2 is shown in Figure 6 as it is not used in the retrieval (see Eq. 4).

Response: Thanks for the comments. We have replaced $BTD_2$ with $BTD_{2-3}$ in Figures 6 and 7, as shown below.

[Figure]

Figure 6: The $BTD_{2-3}$ and $BTD_{1-3}$ for the (a, d) Voronoi, (b, e) Sphere and (c, f) Column ICS models as functions of the IWP and $D_{me}$, respectively.

[Figure]

Figure 7: The difference of $BTD_{2-3}$ and $BTD_{1-3}$ for the (a, c) Voronoi minus Sphere ICS models and (b, d) Voronoi minus Column ICS models as functions of the IWP and $D_{me}$, respectively.

18. Please make Figure 7 easy to understand. For example, indicate the values of IWP and Re with dots ( e.g., Nakajima and King plot). As there are overlapping lines in Figure 7, how cloud properties are retrieved if data fall under such overlapping lines?

Response: According to the comments, we have indicated the values of IWP and Re with dots. When $BTD_{2-3}$ and $BTD_{1-3}$ data fall under such overlapping lines, this overlapping region can lead to obtaining the same IWP and $D_{me}$ when searching the look-up table. That is why the "horizontal line" problem occurs for $D_{me} < 40$ μm and $D_{me} > 40$ μm. This is one of the limitations of our method, which is more applicable for moderate ice particles. And we will improve our retrieval algorithm in the next step.

[Figure]

Figure 7: The LUT of $BTD_{2-3}$ and $BTD_{1-3}$ for the (a) Voronoi, (b) Column and (c) Sphere ICS models varying with the logarithm of IWP and $D_{me}$. Grey dots in circles represent the randomly generated 2000 test data from RSTAR model.

Reference

Brath, M., Fox, S., Eriksson, P., Harlow, C., Burgdorf, M., and Buehler, S.: Retrieval of an ice water path over the ocean from ISMAR and MARSS millimeter and submillimeter brightness temperatures, Atmospheric Measurement Techniques, 11, 611-632, doi:10.5194/amt-11-611-2018, 2018.

Evans, K., Wang, J., Racette, P., Heymsfield, G., and Li, L.: Ice Cloud Retrievals and Analysis with the Compact Scanning Submillimeter Imaging Radiometer and the Cloud Radar System during CRYSTAL FACE, Journal of Applied Meteorology - J APPL METEOROL, 44, 839-859, doi:10.1175/JAM2250.1, 2005.

Evans, K. F., Walter, S. J., Heymsfield, A. J., and McFarquhar, G. M.: Submillimeter-Wave Cloud Ice Radiometer: Simulations of retrieval algorithm performance, Journal of Geophysical Research: Atmospheres, 107, AAC 2-1-AAC 2-21, doi:10.1029/2001JD000709, 2002.

Fox, S., Mendrok, J., Eriksson, P., Ekelund, R., amp, apos, Shea, S., Bower, K., Baran, A., Harlow, C., and Pickering, J.: Airborne validation of radiative transfer modelling of ice clouds at millimetre and sub-millimetre wavelengths, Atmospheric Measurement Techniques, 12, 1599-1617, doi:10.5194/amt-12-1599-2019, 2019.

Heymsfield, A. J., Schmitt, C., and Bansemer, A.: Ice Cloud Particle Size Distributions and Pressure-Dependent Terminal Velocities from In Situ Observations at Temperatures from 0 degrees to-86 degrees C, J Atmos Sci, 70, 4123-4154, 2013.

Marks, C. and Rodgers, C.: A retrieval method for atmospheric composition from limb emission measurements, Journal of Geophysical Research, 981, 10.1029/93JD01195, 1993.

Nakajima, T. and King, M. D.: Determination of the Optical-Thickness and Effective Particle Radius of Clouds from Reflected Solar-Radiation Measurements .1. Theory, J Atmos Sci, 47, 1878-1893, doi:10.1175/1520-0469(1990)047<1878:Dotota>2.0.Co;2, 1990.

Nakajima, T., King, M. D., Spinhirne, J. D., and Radke, L. F.: Determination of the Optical-Thickness and Effective Particle Radius of Clouds from Reflected Solar-Radiation Measurements .2. Marine Stratocumulus Observations, J Atmos Sci, 48, 728-750, doi:10.1175/1520-0469(1991)048<0728:Dotota>2.0.Co;2, 1991.

Nakajima, T. Y. and Nakajima, T.: Wide-area determination of cloud microphysical properties from NOAA AVHRR measurements for FIRE and ASTEX regions, J Atmos Sci, 52, 4043-4059, doi:10.1175/1520-0469(1995)052<4043:Wadocm>2.0.Co;2, 1995.

Nakajima, T. Y., Ishida, H., Nagao, T. M., Hori, M., Letu, H., Higuchi, R., Tamaru, N., Imoto, N., and Yamazaki, A.: Theoretical basis of the algorithms and early phase results of the GCOM-C (Shikisai) SGLI cloud products, Prog Earth Planet Sc, 6, doi:10.1186/s40645-019-0295-9, 2019.

Platnick, S., King, M. D., Ackerman, S. A., Menzel, W. P., Baum, B. A., Riedi, J. C.,

and Frey, R. A.: The MODIS cloud products: algorithms and examples from Terra, IEEE Transactions on Geoscience & Remote Sensing, 41, 459-473, doi:10.1109/TGRS.2002.808301, 2003.

Platnick, S., Meyer, K. G., King, M. D., Wind, G., Amarasinghe, N., Marchant, B., Arnold, G. T., Zhang, Z. B., Hubanks, P. A., Holz, R. E., Yang, P., Ridgway, W. L., and Riedi, J.: The MODIS Cloud Optical and Microphysical Products: Collection 6 Updates and Examples From Terra and Aqua, Ieee T Geosci Remote, 55, 502-525, doi:10.1109/Tgrs.2016.2610522, 2017.

Warren, S. G. and Brandt, R. E.: Optical constants of ice from the ultraviolet to the microwave: A revised compilation, J Geophys Res-Atmos, 113, 2008.

---

## Author Comment (AC4)

Comment on amt-2022-247

Referee comment RC4 on "Retrieval of Terahertz Ice Cloud Properties from airborne measurements based on the irregularly shaped Voronoi ice scattering models" by Ming Li et al.

Anonymous Referee # 1

**General comments**

The paper assessed the capability of the Voronoi and sphere models in the retrieval of IWP and re using aircraft-based measurements of 380, 640, and 874 GHz brightness temperature. Based on the sensitivity analysis, the brightness temperature differences between 640 and 874 GHz are used for IWP retrieval, while brightness temperature differences between 380, 640 and 874 GHz are used for re retrieval. The authors find well correlations between the Voronoi-based retrievals and Evan's Bayesian retrievals using data from the CoSSI instrument. The comparisons of the retrieved IWP and re between Evan's Bayesian retrievals using data from the CoSSIR instrument and the inversion algorithm among the Voronoi and Sphere models suggest that the Voronoi model outperforms the Sphere model. Overall, the highlight of this paper is that the Voronoi model has been previously applied to visible and infrared applications in satellite remote sensing and climate model simulations, and now it is being applied over the terahertz region to investigate how well the model performs there. The paper is relatively wellwritten, and the figures are also well-displayed. The analysis is quantitative and clear, with no obvious flaws. This paper could be a good supplement to the development of satellite remote sensing of ice clouds in the sub-millimetre regions. The topic presented in this study is suitable for Atmospheric Measurement Techniques, so I recommend Minor Revisions for publication.

Response: Thank you very much for your significant comments.

**Specific comments**

1. To ensure the effectiveness and representativeness of the Voronoi model in

terahertz region, I recommend the authors compare the retrievals against more other ice crystal scattering models, such as the column aggregate?

Response: Thanks for the comments. We have added the hexagonal column ice crystal scattering (ICS) model from the ARTS collection of models to compare with our results in this study. The descriptions of the hexagonal column ICS model are added in section 2.1 as shown below.

Lines 149-151: "The randomly-oriented hexagonal column (referred to as the Column hereafter) ICS model was defined by Yang et al. (2000a). Their aspect ratios $a/L$ (defined as the ratio of the semiwidth $a$ of a particle to its length $L$) of Column ICS models are defined as 0.35 and 3.48 respectively when $L$ is less than 100 μm and greater than or equal to 100 μm. The single-scattering property database of the Column ICS model used in the study is developed by Hong (2007, 2009) using the discrete dipole approximation method at frequencies of 100-1000 GHz."

2. How about the accuracy of retrieved IWP and re in previous research in terahertz band? Please illustrate in the introduction.

Response: Thanks for the comments. We have added illustrations about the result accuracy of previous studies in the terahertz band in section 1 as shown below.

Lines 76-81: "For the accuracy of the BMCI method, validation results stated that for clouds with IWP greater than 5 $g/m^2$ the overall median retrieval error is about 30% for IWP and 15% for $D_{me}$ (Evans et al., 2002). Jimenez et al. (2007) used the neural network method to retrieve the IWP and $D_{me}$. Results showed overall median relative errors of around 20% for IWP and 33 μm for $D_{me}$ for a mid-latitude winter scenario, and 17% for IWP and 30 μm for $D_{me}$ for a tropical scenario. Based on these studies, Buehler et al. (2007) proposed a formal scientific mission requirement for a passive submillimeter-wave cloud ice mission based on the background and early research. The requirements are the low IWP should be less than 10 $g/m^2$, the high ice water path should be less than 50%, and the particle diameter should be less than 50 μm. Lately, Liu et al. (2021) proposed an inversion method for the remote sensing of ice clouds at terahertz wavelengths based on a genetic algorithm. Results showed the absolute error of the low IWP (below 20 $g/m^2$) is small, while the relative error of the high IWP is

generally maintained at around 10%, and the absolute error of the effective particle diameter is mostly around 4 μm."

3. Line 98: The "GOA" acronym has not been defined.

Response: We have added the definition of the "GOA" as shown below.

Line 98: ".. several improved Geometrical Optics Approximation (GOA) methods .."

4. I recommend the authors add more analysis and possible explanations in the result section. I recommend the authors relate the single-scattering results to the retrieval results.

Response: Thanks for the comments. In this study, we have added more analysis and explanations of the results in terms of the differences in the single scattering properties of three ice crystal models. The following description is added at the end of section 4.4.

Lines 323-328: "According to the sensitivity results of Figures 6 and 7, the Voronoi ICS scheme has higher $BTD_{2\text{-}3}$ and $BTD_{1\text{-}3}$ compared to the Sphere and Column ICS schemes, especially for large particles and IWP. This characteristic is also shown in Figure 8. This can be explicitly explained by the larger asymmetry factor of the Voronoi ICS model compared to the Sphere and Column ICS models. Thus, stronger forward scattering energy can be detected for the Voronoi ICS model than the other two models. The look-up table of the Voronoi ICS model can cover more IWP and $D_{me}$. The brightness temperature variations of the Voronoi-shaped ice clouds are more prominent and sensitive to the IWP and $D_{me}$. Therefore, the results of the Voronoi ICS model are better than the other two models."

5. In Figure 9, the scattered dots are hard to statistically measure the accuracy. Figure 9 might be better plotted as a PDF of the retrievals.

Response: Thanks. We have modified the scatter plot in Figure 9 to a PDF plot, as shown below.

[Figure]

Figure 9: The joint histogram of differences of (a) the IWP and (b) $D_{me}$ between the retrieved results and the CoSSIR-MCBI algorithm results for the Voronoi (red line), Sphere (black line) and Column models (blue line), separately.

6. Line 117: "Mo.,del" should be "Model".

Response: We have corrected this error as shown below.

Line 117: "the Community Integrated Earth System Model (CIESM)."

Reference:

Buehler, S., Jimenez, C., Evans, K., Eriksson, P., Rydberg, B., Heymsfield, A., Stubenrauch, C., Lohmann, U., Emde, C., John, V., Tr, S., and Davis, C.: A concept for a satellite mission to measure cloud ice water path and ice particle size, Q J Roy Meteor Soc, 133, 109-128, 10.1002/qj.143, 2007.

Evans, K. F., Walter, S. J., Heymsfield, A. J., and McFarquhar, G. M.: Submillimeter-Wave Cloud Ice Radiometer: Simulations of retrieval algorithm performance, Journal of Geophysical Research: Atmospheres, 107, AAC 2-1-AAC 2-21, doi:10.1029/2001JD000709, 2002.

Hong, G.: Parameterization of scattering and absorption properties of nonspherical ice crystals at microwave frequencies, Journal of Geophysical Research, 112, 10.1029/2006JD008364, 2007.

Hong, G., Yang, P., Baum, B. A., Heymsfield, A. J., Weng, F., Liu, Q., Heygster, G., and Buehler, S. A.: Scattering database in the millimeter and submillimeter wave range of 100–1000 GHz for nonspherical ice particles, Journal of Geophysical Research: Atmospheres, 114, doi:10.1029/2008JD010451, 2009.

Jimenez, C., Buehler, S., Rydberg, B., Eriksson, P., and Evans, K.: Performance simulations for a submillimetre wave cloud ice satellite instrument, Q J Roy Meteor Soc, 133, 129-149, 10.1002/qj.134, 2007.

Liu, L., Weng, C., Li, S., Letu, H., Hu, S., and Dong, P.: Passive Remote Sensing of Ice Cloud Properties at Terahertz Wavelengths Based on Genetic Algorithm,

Remote Sensing, 13, 735, 10.3390/rs13040735, 2021.

Yang, P., Liou, K. N., Wyser, K., and Mitchell, D.: Parameterization of the scattering and absorption properties of individual ice crystals, J Geophys Res-Atmos, 105, 4699-4718, 2000a.

---

## Author Comment (AC5)

Comment on amt-2022-247

Anonymous Referee # 5

Referee comment RC5 on "Retrieval of Terahertz Ice Cloud Properties from airborne measurements based on the irregularly shaped Voronoi ice scattering models" by Ming Li et al.

**General comments**

This study compares the capability of the Voronoi and sphere models in the retrieval of IWP and re using aircraft-based terahertz measurements. The study shows that the Voronoi model can provide promising results as compared to Evan's Bayesian retrievals using data from the CoSSI instrument. The inversion algorithm among the Voronoi and Sphere models suggests that the Voronoi model is better than the Sphere model. The paper seems clear and well-written. From the single-scattering properties of ice particles to the ice cloud retrievals, the structure is complete, and the analysis is quantitative. In my opinion, this paper could be a good supplement to the development of ice cloud terahertz remote sensing. The topic presented in this study is suitable for Atmospheric Measurement Techniques. I recommend Minor Revisions for publication.

Response: Thank you very much for your significant comments.

**Specific comments**

1. Are those comparisons between the single-scattering properties of the Voronoi and Sphere models under the same complex refractive index of ice particles? Please add the real and imaginary parts of the refractive index at 325 and 874 GHz in Figures 2 and 4.

Response: According to the comments, we have added the refractive index in the caption of Figures 2 and 4 as shown below.

"Figure 2: The extinction efficiency, single-scattering albedo and asymmetry factor as functions of the SZP for the Voronoi, Column and Sphere ICS models with a refractive index of $1.78 + 0.005i$ in the (a, c, e) 325 GHz and $1.78 + 0.015i$ in the (b, d,

f) 874 GHz frequencies."

"Figure 4: The scattering phase functions for ice particles with four sizes ($R_e$ = 30, 71, 107 and 153 μm) for the (a, d) Voronoi, (b, e) Sphere and (c, f) Column ICS models with a refractive index of 1.78 + 0.005$i$ in the 325 GHz and 1.78 + 0.015$i$ in the 874 GHz, respectively."

2. In the paper, the BTD$_{1\text{-}3}$ may be confused with the BTD$_{1\text{-}2}$-BTD$_{2\text{-}3}$. Please confirm the Acronyms throughout the manuscript.

Response: According to the comments, we have replaced the BTD$_{1\text{-}2}$-BTD$_{2\text{-}3}$ with BTD$_{1\text{-}3}$ and have unified them in the revised manuscript. The relevant descriptions are shown below.

Lines 221-223: "The difference between the 640 GHz BTD and the 874 GHz BTD is simplified to BTD$_{2\text{-}3}$. And the difference between the 380 GHz BTD and the 640 GHz BTD is simplified to BTD$_{1\text{-}2}$. We named the difference between the BTD$_{1\text{-}2}$ and BTD$_{2\text{-}3}$ as BTD$_{1\text{-}3}$."

3. In Figure 6, the brightness temperature differences at 640GHz are shown, albeit not used in the following retrieval. Please give explanations or redraw Figure 6.

Response: According to the comments, we have replaced BTD$_2$ with BTD$_{2\text{-}3}$ in Figures 6 and 7, as shown below.

[Figure]

Figure 6: The BTD$_{2\text{-}3}$ and BTD$_{1\text{-}3}$ for the (a, d) Voronoi, (b, e) Sphere and (c, f) Column ICS models as functions of the IWP and $D_{me}$, respectively.

[Figure]

Figure 7: The difference of $BTD_{2-3}$ and $BTD_{1-3}$ for the (a, c) Voronoi minus Sphere ICS models and (b, d) Voronoi minus Column ICS models as functions of the IWP and $D_{me}$, respectively.

4. I recommend the authors give more possible explanations about why large difference exists for large ice particles in the result section.

Response: We have added reasons on lines 261-264 as shown below.

Lines 273-277: "On the one hand, the higher extinction efficiency and single-scattering albedo of the Voronoi ICS model for large particles are possibly due to the multifaceted shapes of the Voronoi ICS model, which can result in significant side and backward scattering and increase the scattered energy. On the other hand, for large particles, the higher asymmetry factor of the Voronoi ICS model is possibly because the scattered energy is dominated by diffraction. The diffracted energy is concentrated in the forward direction, leading to a large asymmetry factor."